# Previously uncharacterized rectangular bacterial structures in the dolphin mouth

Natasha K. Dudek[1,2,11], Jesus G. Galaz-Montoya [3], Handuo Shi [3,4], Megan Mayer[5,12], Cristina Danita[3], Arianna I. Celis[4], Tobias Viehboeck [6,7], Gong-Her Wu[3], Barry Behr[8], Silvia Bulgheresi[6], Kerwyn Casey Huang [3,4,9], Wah Chiu [3,4,5] & David A. Relman [1,4,9,10] ✉

Much remains to be explored regarding the diversity of uncultured, host-associated microbes. Here, we describe rectangular bacterial structures (RBSs) in the mouths of bottlenose dolphins. DNA staining revealed multiple paired bands within RBSs, suggesting the presence of cells dividing along the longitudinal axis. Cryogenic transmission electron microscopy and tomography showed parallel membrane-bound segments that are likely cells, encapsulated by an S-layer-like periodic surface covering. RBSs displayed unusual pilus-like appendages with bundles of threads splayed at the tips. We present multiple lines of evidence, including genomic DNA sequencing of micromanipulated RBSs, 16S rRNA gene sequencing, and fluorescence in situ hybridization, suggesting that RBSs are bacterial and distinct from the genera *Simonsiella* and *Conchiformibius* (family *Neisseriaceae*), with which they share similar morphology and division patterning. Our findings highlight the diversity of novel microbial forms and lifestyles that await characterization using tools complementary to genomics such as microscopy.

The earliest descriptions of the microbial world centered around morphology and motility patterns of "animalcules"[1]. In the centuries since van Leeuwenhoek's revolutionary discovery, a vast diversity of microbial forms have been described, ranging from star-shaped bacteria in the genus *Stella*[2,3] to the multicellular fruiting bodies characteristic of order Myxobacterales[4,5]. Morphology is a biologically important characteristic, often highly conserved and molded over time by selective pressures resulting from an organism's lifestyle and environmental context[6]. Indeed, cell morphology plays an important role in motility, nutrient acquisition, cell division, and interactions with other cells, including symbioses with hosts, all of which are strong determinants of survival[7]. As such, morphological and structural studies offer an appealing route by which to glean insight into microbial life forms and the mechanisms by which species function and affect their environments. Moreover, characterizing the structures and functions of the diverse range of microbes in uncharted branches of the tree of life provides an opportunity to broaden our understanding of evolution and may result in myriad applications in biotechnology and medicine, exemplified by the development of optogenetics[8] and CRISPR-based gene editing[9].

[1]Department of Medicine, Stanford University School of Medicine, Stanford, CA 94305, USA. [2]Department of Ecology and Evolutionary Biology, University of California, Santa Cruz, Santa Cruz, CA 95064, USA. [3]Department of Bioengineering, Stanford University, Stanford, CA 94305, USA. [4]Department of Microbiology and Immunology, Stanford University School of Medicine, Stanford, CA 94305, USA. [5]Division of CryoEM and Bioimaging, SSRL, SLAC National Accelerator Laboratory, Menlo Park, CA 94025, USA. [6]Department of Functional and Evolutionary Ecology, Environmental Cell Biology Group, University of Vienna, Vienna, Austria. [7]Division of Microbial Ecology, Center for Microbiology and Environmental Systems Science, and Vienna Doctoral School of Ecology and Evolution, University of Vienna, Vienna, Austria. [8]Department of Obstetrics and Gynecology, Stanford University School of Medicine, Stanford, CA 94305, USA. [9]Chan Zuckerberg Biohub, San Francisco, CA 94158, USA. [10]Infectious Diseases Section, Veterans Affairs Palo Alto Health Care System, Palo Alto, CA 94304, USA. [11]Present address: Quantori, Cambridge, MA 02142, USA. [12]Present address: Department of Biological Chemistry and Molecular Pharmacology, Harvard Medical School, Boston, MA 02115, USA. ✉e-mail: relman@stanford.edu

Genomics serves as a powerful lens through which to describe the microbial world. In recent years, metagenomic and single-cell genomic analyses have substantially increased the number of known microbial phylum-level lineages by a factor of nearly four in the bacterial domain[10–13]. Sequencing the genomes of newly discovered organisms has led to the discovery of new functional systems, types of protein variants, and lifestyles[11,14–17], illustrating the correlation between phylogenetic diversity and functional potential[18]. However, the applicability of such approaches is mostly limited to proteins and regulatory systems homologous to those of well-characterized organisms; the prediction of phenotypes and functions that are truly novel and/or whose genetic basis is unknown generally requires complementary knowledge. Given the recalcitrance of most microbial species on Earth to laboratory culturing (in 2016, 72% of approximately phylum-level lineages lacked any cultured representative)[19], methods that do not require cultivated isolates, such as microscopy, offer an appealing, complementary route by which to study novel morphological and functional properties of uncultured lineages. For example, recent advances in cryogenic electron microscopy (cryoEM) and tomography (cryoET) have allowed three-dimensional (3D) imaging of intact bacterial cells at a resolution of a few nanometers[20], leading to important advances in the discovery and characterization of new microbial structures[21,22]. At present, our knowledge of cell biology has been largely limited to observation of and experimentation on bacteria that can be cultured, and thus there exists a severe bias in our understanding toward organisms conducive to growth in laboratory conditions. The use of microscopy and non-sequencing-based techniques for studying "the uncultured majority" will be essential for characterizing the full diversity of bacterial lifestyles that have evolved, particularly efforts at characterizing bacteria in relatively unstudied environments, such as the oral cavity of bottlenose dolphins (*Tursiops truncatus*), that host a rich collection of novel microbes and functional potential[16,23].

Despite the diversity of microbial cell shapes, rectangular structures are a rarity, with a poorly understood genetic basis. Such structures are of two types: individual cells that are rectangular and cell aggregates that form rectangles. To the best of our knowledge, the discovery of non-eukaryotic rectangular cells has thus far been restricted to the family *Halobacteriaceae*, which consists of halophilic Archaea. Known rectangular cells from this family include *Haloquadratum walsbyi*[24], *Haloarcula quadrata*[25], and members of the pleomorphic genus *Natronrubrum*[26]. FtsZ-based fission has recently been observed in the cube-shaped nematode symbiont *Candidatus* Thiosymbion cuboideus[27] and additional rectangular cells believed to be bacterial or archaeal have been discovered in high salinity environments but not taxonomically identified[28]. Among eukaryotic microorganisms, diatoms can have a rectangular appearance when visualized in two dimensions, although these cells are cylindrical rather than rectangular prisms[29]. A variety of bacteria form rectangular cell clusters, such as sheets of coccoid bacteria (e.g., *Thiopedia rosea* and the genus *Merismopedia*[30]), cuboidal structures of coccoid bacteria (e.g., the genera *Sarcina* and *Eucapsis*[30]), and rectangular trichomes formed by disc-shaped bacteria (e.g., *Oscillatoria limosa* and other cyanobacteria[30]).

Also rare in the microbial world are cells that diverge from the typical pattern of cell division along a transverse axis. Two spectacular examples are *Candidatus* Thiosymbion oneisti and *Candidatus* Thiosymbion hypermnestrae[31–33], which have been exclusively found on the surface of marine nematodes. This division patterning is thought to preserve attachment to the host[31,32]. Similarly, members of the family *Neisseriaceae* genera *Alysiella*, *Simonsiella*, and *Conchiformibius* divide longitudinally, which is thought to help with adherence to human epithelial cells in the oral cavity[34]. These taxa are further striking in that they can be regarded as multicellular bacteria. Additional examples of bacteria that undergo longitudinal division include *Spiroplasma*

poulsonii[35] and genus *Candidatus* Kentron, a clade of symbionts hosted by the marine ciliate *Kentrophoros*[36,37]. Such insight into the reproductive methods of diverse bacteria is essential for building a comprehensive understanding of cell biology.

Here, we discover unusual rectangular bacterial structures (RBSs) in dolphin oral samples and characterize their cellular dimensions and DNA patterning using phase-contrast and fluorescence microscopy. Regular bands of DNA suggest that the units are sheets of individual cells, each of which is encapsulated by an inner and outer membrane, while fluorescence in situ hybridization (FISH) experiments and metagenomic sequencing strongly suggest that they are bacterial and potential members of the class Betaproteobacteria. Using cryogenic transmission electron microscopy (cryoTEM) and cryoET, we characterize the envelope structure of RBSs and discover previously unobserved surface features such as heterogeneous bundles of appendages that protrude from the ends of RBSs and splay out at the tips. These findings highlight the power of high-resolution microscopy for exploring the nature of uncultivated microbes.

## Results

### Rectangular bacterial structures in the dolphin oral cavity are Gram-negative and contain multiple bands of DNA

Oral swab samples were collected from the mouths of eight bottlenose dolphins (*Tursiops truncatus*) under the purview of the U.S. Navy Marine Mammal Program (MMP) in San Diego Bay, California, USA, during three distinct intervals in 2012, 2018, and 2022 (Methods). RBSs were readily apparent in phase contrast microscopy images (Fig. 1a–e). The RBSs resembled rectangular prisms; they were not cylindrical (Supplementary Movie 1). They exhibited Gram-negative characteristics following Gram staining (Fig. 1f). Attempts at membrane staining using FM4-64 were unsuccessful, as the FM4-64 dye did not stain any part of the RBSs. RBSs contained multiple parallel bands of fluorescence with DAPI staining (Fig. 1b–e). In some RBSs, the neighboring DNA band pairs appeared "H"-like (Fig. 1g, white arrow), suggesting two rod-shaped cells in the process of division along a longitudinal axis with the DNA bands undergoing segregation[27]. RBSs clustered into two morphotypes based on the dimensions of the rectangular units (Fig. 1h). Using materials from a single dolphin oral swab sample, we quantified the length and width of 23 RBSs. Morphotype A exhibited a median length of $3.95 \pm 2.89$ μm median absolute deviation (MAD) and median width of $5.08 \pm 0.10$ μm MAD ($n = 15$ RBSs). Morphotype B had a median length of $3.08 \pm 0.93$ μm MAD and a median width of $2.21 \pm 0.56$ μm MAD ($n = 8$ RBSs). The dimensions for both length and width were significantly different between the two morphotypes (for length, $p = 0.02$; for width, $p = 10^{-10}$; two-sided Welch's $t$-test). The different morphotypes may represent different cell types or taxonomic groups (e.g., strains or species), cells in different stages of development, or cells with altered shape in response to environmental conditions.

We assessed the prevalence of RBSs of each morphotype in a set of 73 oral swab samples collected from eight dolphins (Supplementary Data 1) using a high-throughput, automated phase contrast microscopy workflow that collected imaging data for 226 fields of view per sample[38]. RBSs of morphotype A, henceforth referred to as RBS-As, were detected in 39/73 samples from 7/8 dolphins (note that the number of samples per dolphin is not constant and that different oral locations were swabbed in different years). RBSs of morphotype B, henceforth referred to as RBS-Bs, were detected in 42/73 samples from 6/8 dolphins. Of 25 of these 73 samples that were collected from distinct oral sites, RBSs were detected in palatal (RBS-A = 5/5; RBS-B = 4/5) and gingival samples (RBS-A = 12/15; RBS-B = 14/15), but less frequently in buccal samples (RBS-A = 0/5; RBS-B = 2/5). Thus, we infer that the RBSs are stable colonizers in the dolphin oral cavity and have preferred colonization locations.

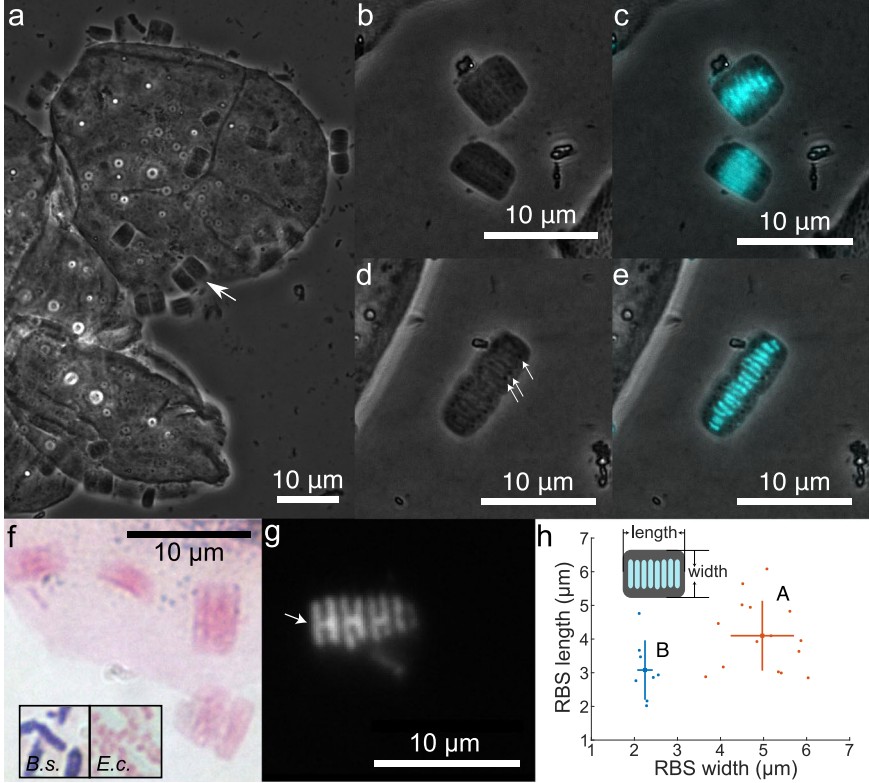

**Fig. 1 | Light microscopy reveals RBSs with multiple morphotypes and distinct DNA banding patterns. a** Phase-contrast images of RBSs (arrow indicates one example) on the surface of dolphin oral epithelial cells. Also see Supplementary Movie 1. **b, c** Some RBSs exhibited long bands of DAPI fluorescence. Phase-contrast image is shown in (**b**), with fluorescence overlay in cyan in (**c**). **d, e** Other RBSs exhibited shorter DAPI bands. Phase-contrast image is shown in (**d**), with fluorescence overlay in cyan in (**e**). Dark spots (arrows) were organized in lines perpendicular to DAPI-stained bands. DAPI-stained bands appeared to be organized in pairs. **f** Gram-stained RBSs display Gram-negative characteristics. Inset: Gram-stained *Bacillus subtilis* (*B.s.*, Gram-positive) and *Escherichia coli* (*E.c.*, Gram-negative). **g** Neighboring DNA band pairs in an RBS form "H"-like shapes (arrow), likely because the DNA bands are segregated nucleoids in a cell undergoing longitudinal division. **h** The two RBS morphotypes have distinct distributions of length and width. The median width and length of 15 RBS-As measured $3.95 \pm 2.89\,\mu m$ MAD and $5.08 \pm 0.10\,\mu m$ MAD, respectively, and for 8 RBS-Bs measured $3.08 \pm 0.93\,\mu m$ MAD and $2.21 \pm 0.56\,\mu m$ MAD, respectively. The centers of the orange and blue crosses represent the means of RBS-As and RBS-Bs, respectively, while the lengths of the arms represent ±1 standard deviation.

In this study, we focused on RBS-A, as this morphotype had a higher abundance in the dolphin oral samples and is more morphologically distinct from other known microbes.

### RBSs are likely bacterial and not affiliated with *Neisseriaceae*

Given the intriguing morphology of RBSs, we next sought to determine their taxonomic affiliation. RBSs morphologically resemble members of the genera *Simonsiella* and *Conchiformibius* (family *Neisseriaceae*), which are rod-shaped oral commensals in mammals[39]. In a re-analysis of the Sanger clone library and 454 pyrosequencing data from a previous 16S rRNA gene amplicon survey of gingival swab samples from 38 dolphins from the same population[23], no *S. muelleri* (the sole species of the genus *Simonsiella*) or genus *Conchiformibius* amplicons were detected in any of these samples, although other members of the family *Neisseriaceae* were detected in these dolphin oral samples. A putative *S. muelleri* sequence type was detected in the Sanger library from the mouth and gastric fluid of one sea lion examined in the same amplicon study[23] (NCBI accession number JQ205404.1); this partial 16S rRNA gene sequence has 94.6% identity over 99% query coverage to the partial *S. muelleri* ATCC 29453 16S rRNA gene sequence (NCBI accession number NR_025144.1). The former study detected five other family *Neisseriaceae* sequence types in the Sanger library from sea lions (mouth and stomach), water, and fish species fed to marine mammals[23]. Meanwhile, there were four family *Neisseriaceae* sequence types recovered in the 454 pyrosequencing survey, from dolphins (stomach, respiratory system), sea lions (mouth, stomach), fish, and seawater. These positive identifications indicate that *S. muelleri* and

family *Neisseriaceae* DNA could be extracted successfully in the previous study, although they were not detected in any dolphin oral samples[23].

We then performed 16S rRNA gene amplicon sequencing on 54 dolphin oral samples (Methods), resulting in the detection of 1,116,394 amplified sequence variants (ASVs) from 5,339,751 sequence reads. Of these 54 samples, 48 were screened in a high-throughput manner for RBSs via phase contrast microscopy (Methods). RBS-As and RBS-Bs were each detected in 22/48 samples (45.8%), but not the same 22 samples. One or the other morphotype was detected in 30/48 samples (62.5%). Rarefaction curves suggested that the depth of sequencing was sufficient for results to be near saturation for ASV richness (Fig. 2a). No family *Neisseriaceae* amplicons were detected in these samples (Fig. 2b), despite our ability to recover *S. muelleri* sequences from previously negative oral swab samples after deliberately spiking aliquots of these samples with *S. muelleri*. ASVs common to the RBS-A and RSB-B positive samples can be seen in Supplementary Tables 1 and 2, respectively.

The marine origin and rectangular nature of the RBSs also gave rise to speculation that they may be marine diatoms (e.g., *Skeletonema costatum*), as cylindrical marine diatoms can appear rectangular in two dimensions. Thus, we next evaluated a potential eukaryotic origin of RBSs. We performed FISH using labeled eukaryotic (Euk-1209) and bacterial (Eub-338) probes, the latter of which is known to hybridize with both bacteria and archaea. As controls, we cultured and included *S. costatum* and the bacterium *Escherichia coli*. The Eub-338 probe hybridized to both *E. coli* and the RBSs, while the Euk-1209 eukaryotic

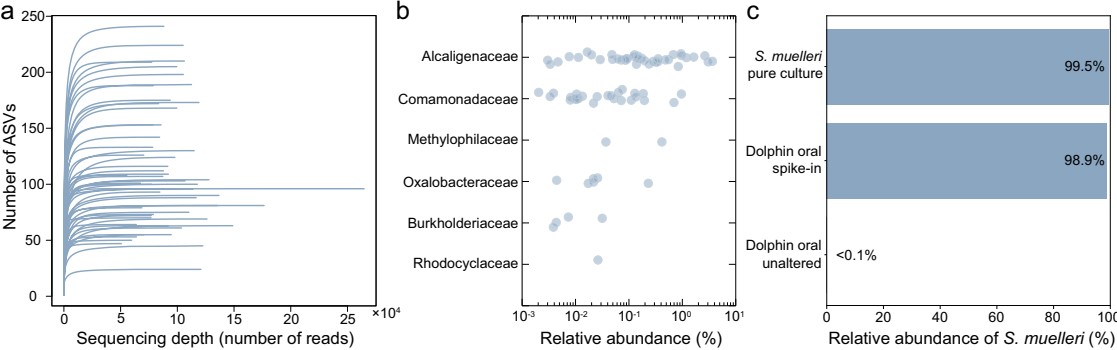

**Fig. 2 | 16S rRNA gene amplicon sequencing of dolphin oral samples indicates that RBSs are not affiliated with the class Betaproteobacteria family *Neisseriaceae*.** Dolphin oral samples (*n* = 54) were subjected to deep 16S rRNA gene amplicon sequencing. **a** Rarefaction curves for the 54 sequenced dolphin oral samples. **b** For each family of class Betaproteobacteria detected in the 54 dolphin oral samples, the relative abundance of member ASVs is plotted. Note that the genus *Simonsiella* is a member of the class Betaproteobacteria family *Neisseriaceae*; no ASVs affiliated with this family were detected. Visual examination using phase-contrast microscopy (see Methods) revealed RBSs of either morphotype A or B in 39 of 73 samples screened (note that in total 73 samples were visually screened for RBSs; 54 were used for amplicon sequencing, while the remainder were used for other experiments) (Supplementary Data 1). **c** Sequencing of an *S. muelleri* pure culture and a dolphin oral sample (confirmed to contain RBS-As) with *S. muelleri* spiked resulted in the detection of a family *Neisseriaceae* ASV. The same ASV was detected at a relative abundance of <0.1% in that same dolphin oral sample in its unaltered state (no *S. muelleri* added) when prepared and sequenced in parallel with the *S. muelleri* positive samples; the ASV was not detected in any dolphin oral sample prior to the introduction of *S. muelleri* into the lab environment.

probe hybridized to *S. costatum* cells alone (Supplementary Fig. 1), indicating that RBSs are not eukaryotic and thus not diatoms.

With no further a priori hypotheses as to the specific nature of the RBSs, we pursued a variety of general approaches to shed light on their identity. First, we cultured oral samples under aerobic and anaerobic conditions in three media used to grow diverse bacteria (Methods), hoping to enrich for RBSs. Unfortunately, RBSs were not visible upon inspection of cultures under a microscope, indicating that the growth requirements for RBSs are distinct from those of typical bacteria isolated from mammalian microbiota.

We further explored the potential for culturing RBSs directly on solid surfaces. Among the swab samples from 2022, we stored five in 20% glycerol immediately after collection to maximize the chances of maintaining RBS viability. Single-cell microscopy identified two glycerol stocks containing samples with numerous RBSs. These stocks were inoculated on agar plates containing either BSTSY-FBS or BHI-blood medium (Methods). BSTSY-FBS plates enable the growth of *S. muelleri* and were incubated aerobically at 37 °C. BHI-blood plates are typically used to grow a variety of bacterial commensals from mammals; these plates were incubated anaerobically at 37 °C. After 3 weeks of extended incubation, no colonies were visible on the BSTSY-FBS plates, while a control sample of *S. muelleri* formed large colonies after 1–2 days. The BHI-blood plates collectively contained ~300 colonies, of which we examined 288 using high-throughput single-cell microscopy[38]. None of the colonies contained cells with morphology similar to RBSs. Taken together, while disappointing from the point of view of enabling genomics approaches, our inability to culture RBSs provides further support that they are not *S. muelleri*, which is readily culturable using such approaches.

Our next strategy employed mini-metagenomics, an approach in which a small number of cells are subsampled from a complex community and their DNA is amplified using multiple displacement amplification (MDA). Notably, this approach largely avoids preconceived biases about possible identity, since metagenomic analyses should reveal DNA from any cell from any domain of life, assuming successful cell lysis. We used three techniques to capture RBS-As for genomic sequencing: laser capture microdissection, microfluidics, and cell micromanipulation. Due to their large size compared with other bacteria, low density, and the propensity of RBS-As to stick to other cells and to abiotic surfaces, only micromanipulation led to successful RBS-A capture (Supplementary Fig. 1). In addition to four collection

tubes, each containing ~1–3 RBS-As, four negative-control tubes of sample fluid were collected with the micropipette without any cells visible at the resolution of our microscope. Cell-free DNA and small non-target cells were also likely collected along with the RBS-As, given the frequent close proximity of RBS-As to other cells. DNA from RBS-A-positive and RBS-negative samples was amplified using MDA, co-assembled, and sorted into 18 genome bins (Supplementary Fig. 3 and Supplementary Tables 3 and 4; Methods). Notably, no *S. muelleri*, family *Neisseriaceae*, or diatom genomes were recovered, although positive controls for these taxa were not included in the experimental design. The eukaryotic bins matched the human genome or the fungal class Malasseziomycetes, members of which are known commensals of human skin[40], and hence may represent contaminants. No archaeal bins were recovered.

As half of the candidate bacterial bins recovered from the mini-metagenomics experiment were members of the phylum Proteobacteria, we next performed FISH experiments using a set of class-specific probes targeting Alpha-, Beta-, and Gammaproteobacteria. The RBSs exhibited positive binding only to class Betaproteobacteria probes (Fig. 3).

We synthesized insights into RBS-A taxonomic identity from each experimental line of evidence (Fig. 4). While no conclusive insights can be drawn about RBS-A identity, FISH results suggested an affiliation with class Betaproteobacteria, and a single partial genome from class Betaproteobacteria was recovered from the mini-metagenomics experiments; the affiliation of this bin (number 16) with family *Alcaligenaceae* was confirmed via phylogenetic analysis of the 16S rRNA and ribosomal protein S3 genes (Supplementary Figs. 4 and 5 and Methods). An ASV consistent with the 16S rRNA gene of that bin is present in 11/13 of samples in which RBS-As were visually confirmed to be present and for which amplicon sequencing data are available.

The evolutionary path toward multicellularity and longitudinal division is poorly understood. Recent efforts to identify the genetic basis of these bacterial characteristics in the family *Neisseriaceae* found that the acquisition of the amidase-encoding gene *amiC2*, along with modifications to key regulatory genes (e.g., *mreB*, *ftsA*), likely plays an important role. We searched the bins recovered from the mini-metagenomics experiment for putative AmiC2 proteins (those encoding Amidase_3, PF01520); candidates were identified in bins 4, 7, 9, 10, 11, 12, and 16 (the last of which is affiliated with *Alcaligenaceae*). Upon confirmation of the identity of the RBSs, future work should

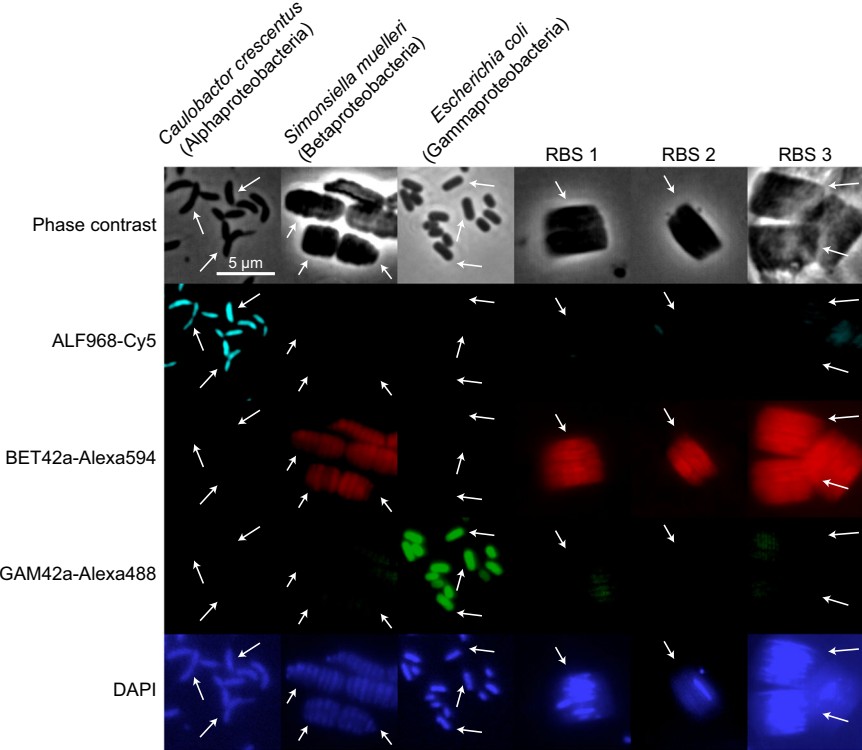

**Fig. 3 | Fluorescence in situ hybridization indicates that RBS-As are affiliated with class Betaproteobacteria.** Probes for phylum Proteobacteria classes Alphaproteobacteria (ALF968), Betaproteobacteria (BET42a), and Gammaproteobacteria (GAM42a) were assessed for their ability to hybridize to RBSs. Top row: phase-contrast images; middle three rows from top to bottom: fluorescence images for class Alphaproteobacteria, Betaproteobacteria, and Gammaproteobacteria probes, respectively; bottom row: DAPI staining of presumed DNA. Arrows highlight relevant cells in samples. RBSs of the "A" morphotype (Fig. 1h) hybridized with the class Betaproteobacteria probe and exhibited minimal hybridization with the class Alphaproteobacteria and Gammaproteobacteria probes.

| Bin | Phylum | Lowest taxonomic ID | RBS-A 1 | 2 | 3 | 4 | Neg 1 | 2 | 3 | 4 | Genomics | Gram stain | 16S 75%+ | FISH |
|---|---|---|---|---|---|---|---|---|---|---|---|---|---|---|
| 16 | Betaproteobacteria | f_Alcaligenaceae | | | | | | | | | | | 11 | |
| 4 | Epsilonproteobacteria | g_Arcobacter | | | | | | | | | | | 13 | - |
| 3 | Epsilonproteobacteria | f_Campylobacteraceae* | | | | | | | | | | | 13, 13 | - |
| 18 | Gammaproteobacteria | p_Gammaproteobacteria | | | | | | | | | | | N/A | |
| 11 | Bacteroidetes | g_Tenacibaculum* | | | | | | | | | | | 12, 12, 10 | |
| 15 | Gammaproteobacteria | p_Gammaproteobacteria | | | | | | | | | | | N/A | |
| 13 | Gammaproteobacteria | f_Moraxellaceae* | | | | | | | | | | | 13, 13, 10 | |
| 17 | Gammaproteobacteria | g_Pasteurella* | | | | | | | | | | | 13, 13 | |
| 12 | Bacteroidetes | f_Flavobacteriaceae | | | | | | | | | | | 13 | - |
| 6 | Fusobacteria | g_Fusobacterium | | | | | | | | | | | 13 | - |
| 10 | Gammaproteobacteria | p_Gammaproteobacteria | | | | | | | | | | | N/A | |
| 9 | Bacteroidetes | g_Porphyromonas* | | | | | | | | | | | 11 | - |
| 14 | Fusobacteria | g_Oceanivirga | | | | | | | | | | | | - |
| 5 | Gracilibacteria | p_Gracilibacteria | | | | | | | | | | | † | 10 | - |
| 2 | Gracilibacteria | p_Gracilibacteria | | | | | | | | | | | † | 10 | - |
| 7 | Actinobacteria | f_Corynebacteriaceae* | | | | | | | | | | | | | - |
| 8 | Fungi | c_Malasseziomycetes | | | | | | | | | | | | - | |
| 1 | Metazoa | s_Human | | | | | | | | | | | | - | |

**Fig. 4 | Insights into RBS-A identity.** The table shows the 18 bins recovered from MDA and sequencing of RBS-A samples collected via micromanipulator, along with the phylum from which they are inferred to derive. For the lowest taxonomic identity achieved (or class, in the case of the polyphyletic phylum Proteobacteria), an asterisk (*) denotes lower confidence in the assignment (Methods). The RBS-A panel depicts relative abundances of bins in each of the four samples that contained RBS-As based on visualization, color-coded as follows: green: ≥5%, yellow: ≥1% and <5%, orange: >0% and <1%, red: 0%. The negative-control panel (Neg) presents the same information for each of the samples that did not appear to contain RBS-As. Criteria for gauging the likelihood of a bin deriving from the RBS-As are shown: (Genomics) Was the bin ever present in negative controls (green = no, yellow = yes but never ≥1% relative abundance, orange = yes but never ≥5%, red = yes and at least once ≥5%)? (Gram stain) Are members of this taxonomic group known to be Gram-negative, like the RBSs? (16S 75%+) Of ASVs detected in >75% of samples that underwent 16S rRNA gene amplicon sequencing and were visually confirmed to contain >10 RBS-As (n = 13), was an ASV of this taxonomic identity? Numbers in boxes indicate the number of samples in which this ASV was present. For recovered bins detected only to the level of class Gammaproteobacteria, a large and diverse group, this criterion is marked in yellow and ASV counts are not shown. (FISH) Based on FISH results, which bins are supported as potential candidates for the RBS-As? A cross (†) denotes that members of the phylum Gracilibacteria are inferred not to be Gram-negative from genomic studies (although they are not necessarily Gram-positive)[92]. The highest likelihood candidates are green for all three criteria.

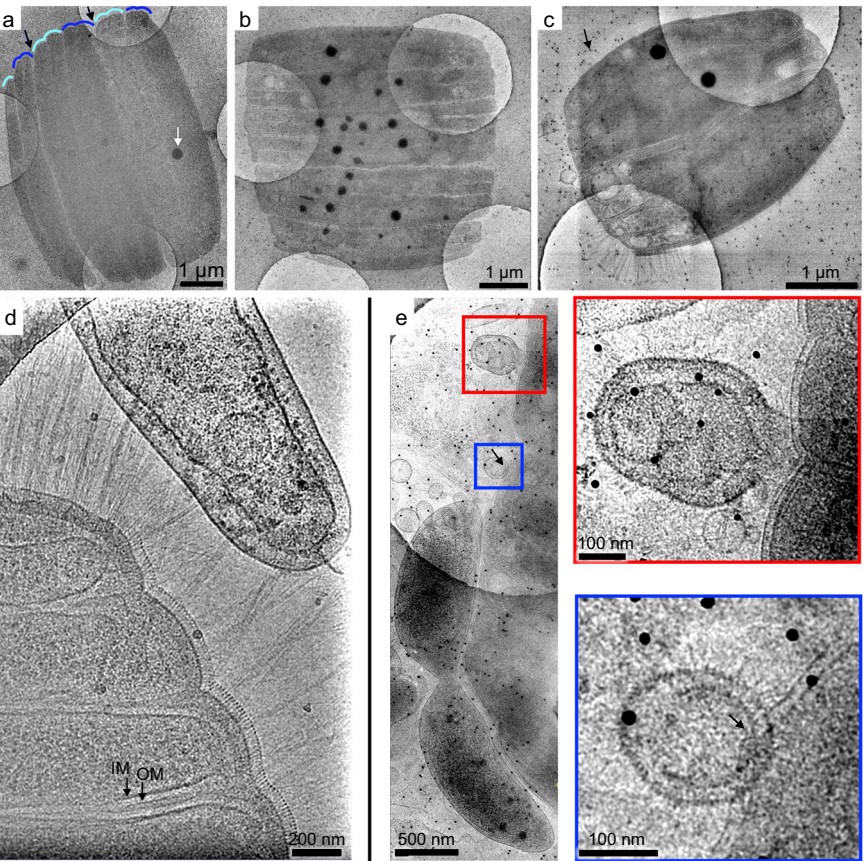

**Fig. 5 | CryoTEM demonstrates that RBSs consist of multiple parallel, likely paired segments and are often near other cells.** Band-pass filtered and denoised low-magnification **a**, **b** images or **c** montage cryoTEM images of RBSs on an R2/2 holey carbon TEM grid. Higher-magnification images showing **d** pilus-like appendages and **e** proximal cells and interacting vesicles at the RBS periphery. In (**a**), pairs of segments are highlighted with alternating shades of blue, and sharp indentations between groups of segments are denoted with black arrows. Dense spheroidal objects were present inside RBSs (white arrow). In (**d**), segments are encapsulated by an inner membrane (IM) and outer membrane (OM), denoted by black arrows. **d**, **e** Representative micrographs showing that RBSs were often in close physical proximity to other cells in the samples. In (**e**), an apparent small indentation (black arrow) in the RBS periodic surface covering overlaps with a non-RBS cell or vesicle. Small dark spots (15 nm) in (**c**–**e**) are gold fiducial particles used for tilt series alignment in cryoET experiments (black arrow).

examine the significance of *amiC2* in conferring their unusual morphology.

## CryoEM and cryoET reveal nanoscale surface and internal structures of RBS-As

To gain high-resolution structural insight into RBS-As, we imaged dolphin oral samples containing high densities of RBS-As using cryo-TEM. Low-magnification cryoTEM images revealed that each RBS-A consists of seemingly paired segments organized in parallel (Fig. 5a, b). These segments were oriented similar to the DAPI-stained bands seen in fluorescence microscopy images (Fig. 1c, e). Some groups of segments appeared to be in the act of separating from other groups, although our static data cannot definitively say whether these observations were reflective of cell division. Segments were surrounded by a dense, membrane-like layer under a low-density layer (Fig. 6a, b (right) and Fig. 7a; Supplementary Movies 2 and 3).

We hypothesized that RBSs are most likely aggregates of cells, with each DNA-containing segment corresponding to an individual cell. The following observations support this hypothesis: (1) each individual segment appeared to be surrounded by an inner and outer membrane, reminiscent of the plasma and outer membrane seen in other Gram-negative bacteria (Figs. 5a, 6a, b (right) and 7a); (2) segments are arranged in the same geometry (Fig. 5a, b) as the DAPI-stained bands and FISH probe-hybridized bands (Figs. 1c, e, g and 3); (3) appendages protruded from the surface of individual segments (Fig. 5c, d); (4) RBS-As often consisted of variable numbers of segments that appeared to be separating from one another (Figs. 1d, e and 6d, e), suggesting that the rectangular structures do not reflect individual cells; (5) neighboring segments appeared H-shaped (Fig. 1g), reminiscent of nucleoids segregating in a bacterial cell undergoing longitudinal division; and (6) while a recent report describes the first discovery of a bacterium in which DNA is stored in a membrane-bound compartment[41], the number of known bacterial species in which DNA is physically segregated from the cytoplasm, let alone organelle-bound, is extremely low. By contrast, multicellularity has been documented in diverse bacterial species (reviewed in ref. 42).

Dark, spherical structures were visible in the body of RBSs in cryoTEM images (Fig. 5c). In one tomogram, two dense spheroidal objects were prominently visible and measured 192 nm × 200 nm × 192 nm (volume $3.1 \times 10^7$ nm$^3$) and 215 nm × 220 nm × 220 nm (volume $4.4 \times 10^7$ nm$^3$). Vesicle-like structures were also apparent. Notably, a surface covering with a periodicity of ~7–9 nm encapsulated the RBS-As (mean $7.83 \pm 0.86$ nm SD) (Figs. 6 and 7 and Supplementary Movies 2 and 3). This feature was measured from the high-resolution 2D micrographs of two RBS-As from a single dolphin oral sample by generating line-density plots of segments (2 from each image) and manually measuring the distance between intensity peaks for 6–7 peaks (Fig. 7d, e).

To obtain more detailed 3D reconstructions of RBS-A features, we conducted cryoET experiments. Tilt-series acquisitions were limited to the RBS-A periphery since the thickness of the RBS-A bodies occluded the electron beam almost completely at high tilt angles. The thickness

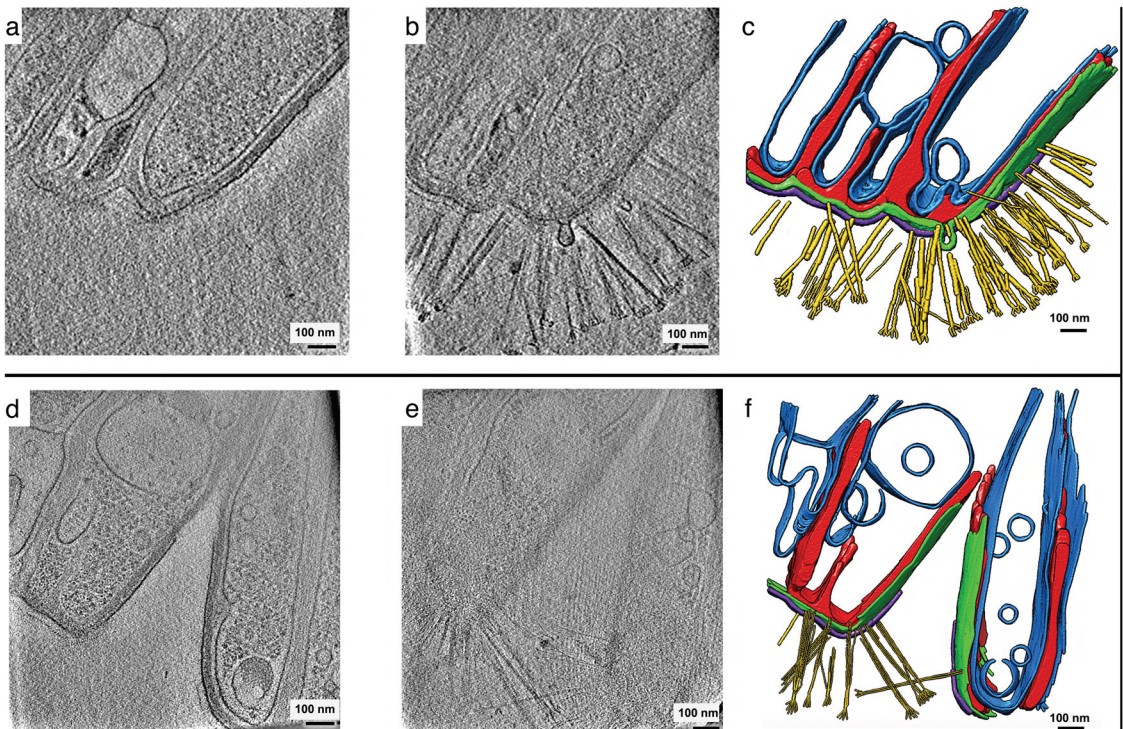

**Fig. 6 | CryoET reveals the three-dimensional architecture of RBS-A components. a, b, d, e** Examples of ~3-nm-thick slices at two different depths (**a** versus **b, d** versus **e**) from each of two representative RBS-A tomograms. **c, f** Corresponding manual annotations of cellular features. The tomograms are thick (~500–600 nm) and as such, pilus-like appendages (yellow: **c, f**) were visible only at a certain depth within the tomographic volume density of the RBS-A. Also see Supplementary Movies 2 and 3. **c, f** Blue, inner membranes; purple, periodic surface covering; green, outer membrane; red, matrix. Scale bars, 100 nm.

at the RBS-A periphery (<1 μm from the edge) ranged from ~323–751 nm, with an average value of ~509 ± 132.4 nm SD ($n = 15$), (above the ~500 nm threshold commonly regarded as the upper limit for productive cryoET imaging[43]). Attempts to image the RBS-A body with cryoET failed because gold fiducials and cellular features quickly became indiscernible upon tilting, suggesting that the thickness of the specimen induced too many multiple scattering events[44].

Appendages that resembled pili[45,46] protruded from RBS-As; these appendages often consisted of hair-like structures that formed bundles and splayed out at the tips, sometimes intertwining and/or crossing over one another (Figs. 6b and 7b, c). The bundles of appendages were structurally heterogeneous, with variable lengths, bundle widths, and numbers of tips. Notably, in examining the various features within the tomograms, we did not observe any membrane-bound organelles reminiscent of a nucleus, in line with a non-eukaryotic identity.

For both the appendages and periodic surface covering, sub-tomogram averaging[47] did not yield consistent maps, likely due to the thickness of the RBS-A periphery (often >500–600 nm thick), low signal-to-noise ratio of the tomograms, and limiting characteristics of the features in question, such as the variable curvature of the regions with stretches of continuous periodic surface covering. Successful subtomogram averaging typically relies on averaging identical structures, for example, repeated copies of a macromolecular complex, such as ribosomes in the same functional state. One can compensate for structural variability in the form of conformational and compositional heterogeneity with large datasets composed of thousands of subtomograms in combination with advanced classification methods. However, in our datasets, both the pilus-like appendages (Fig. 7a–c) and the S-layer-like surface feature (Fig. 7a, d, e) were structurally heterogenous and sparsely distributed in the RBS-As (e.g., the S-layer-like surface feature is not continuous along the entire membrane of the RBS-As, and in the stretches where it is, it exhibits differential curvature) and thus were observed only in a fraction of our tomograms.

To address sample thickness, we used cryogenic focused ion beam (FIB) scanning electron microscopy (SEM)[48] to mill thinner lamellae of RBS samples[49]. However, we could not locate any RBS-As under the ice, for two possible reasons: (1) with an inferred thickness between ~0.6 and 1.7 μm, RBS-As may not form "mounds" under the ice that are protuberant enough to suggest where to mill; and (2) other larger cells in the samples (such as dolphin epithelial cells) formed more prominent mounds that obscured the RBS-As. Indeed, none of the lamella we produced from candidate locations contained RBS-As. The data from this experiment suggest that cryo-correlative light and electron microscopy (cryoCLEM) will be necessary in future studies to locate candidate regions in cryoEM grids with vitrified RBS-As from which to produce thin lamellae using cryoFIB-SEM. Nonetheless, the cell surface exhibited features similar to those we observed on the RBS-A periphery, namely pili and S-layers. We suggest that the pilus-like appendages and the S-layer-like periodic surface covering of RBS-As merit future investigation.

## Discussion

Here, we used optical microscopy, cryoTEM, and cryoET to search for novel morphological diversity within the microbiota of dolphin oral samples. Morphological diversity was predicted based on previous findings of novel phylogenetic diversity and functional potential in the dolphin mouth via sequencing-based studies[16,23]. Interestingly, we discovered unusual RBSs. We infer that both RBS morphotypes are indigenous to the dolphin mouth given that they were consistently present in this environment: RBS-As and RBS-Bs were identified in 39/73 and 42/73 samples, respectively, that were surveyed using high-throughput microscopy imaging, including in 7/8 and 6/8 dolphins included in this study, and were present in samples collected during intervals ten years apart. Previous studies have found that the

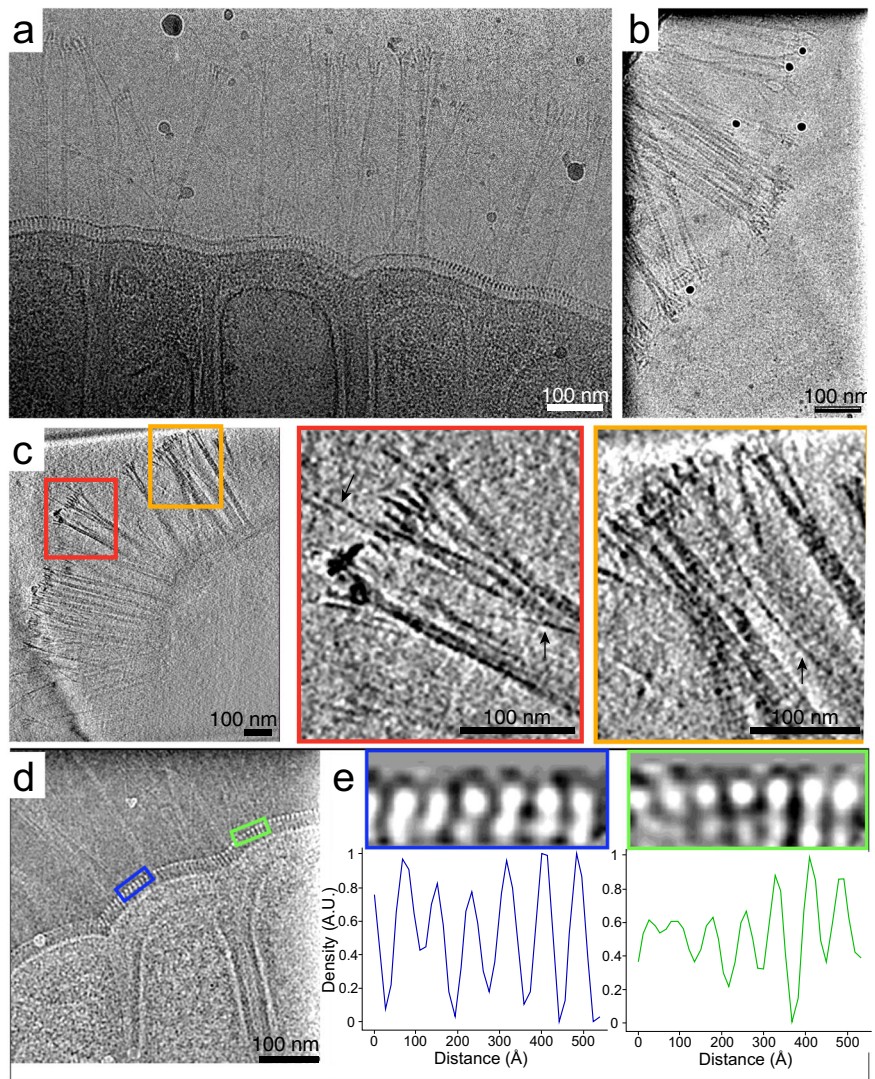

**Fig. 7 | RBS-A surface features include appendages in heterogeneous bundles that splay out at the tips and a periodic surface covering around the entire RBS-A. a** Single cryoTEM image. **b**, **c** CryoET slices (~3-nm thick) of an RBS-A. Red and orange boxes in (**c**) are magnified views of bundles of appendages, and arrows denote thin, single appendages. **d** Representative 2D cryoTEM image at the edge of an RBS-A showing a periodic surface covering. **e** Line-density profiles along selected regions from the image in (**d**) show that the spacing of the repetitive features is ~7–9 nm along a direction parallel to the RBS-A membrane.

microbiota of marine mammals is distinct from that of seawater (even that of skin, which is constantly in direct contact with seawater)[23,50], and thus it is unlikely that RBSs are simply contaminants from seawater.

The taxonomic identification of specific cell morphotypes from complex communities can be extremely difficult, to the point that it often remains unresolved[28,51,52]. Results collected here strongly suggest that the RBS-As are not affiliated with the multicellular longitudinally dividing family *Neisseriaceae* members, *S. muelleri*, genus *Conchiformibius*, or genus *Alysiella*, that can also form rectangularly shaped clusters of rod-like cells[53]. First, the marker gene amplicon sequencing workflow employed here did not detect any family *Neisseriaceae* amplicons in 54 dolphin oral samples that underwent deep 16S rRNA gene amplicon sequencing, although this workflow did detect *S. muelleri* after cells from this taxon were intentionally spiked into aliquots of dolphin oral samples. Second, attempts to culture RBS-As using techniques that were employed successfully in our laboratory to culture *S. muelleri* failed, suggesting that the RBS-As have different physiological requirements for growth than those required by *S. muelleri*. Third, no family *Neisseriaceae* genomes were recovered from the mini-metagenomics experiment. Fourth, visual comparisons of

RBS-A images presented here with the TEM and fluorescence microscopy images of multicellular longitudinally dividing family *Neisseriaceae* (genera *Alysiella*, *Simonsiella*, and *Conchiformibius*) in ref. 53 suggest that they are different taxa. For example, RBSs appear to contain segments (cells) that are embedded in a matrix-like material and fully encapsulated by an S-layer-like structure, whereas multicellular longitudinally dividing family *Neisseriaceae* belonging to the same filament only appeared to share their outer membrane[53].

FISH experiments indicated that the RBS-As are bacterial and likely members of the class Betaproteobacteria. A class Betaproteobacteria genome was recovered from the mini-metagenomics experiment from the family *Alcaligenaceae*. An ASV matching the 16S rRNA gene of this bin was detected in 11/13 samples that underwent amplicon sequencing and were visually confirmed to contain RBS-As via phase contrast microscopy. Taken together, the RBS-As may be members of the family *Alcaligenaceae*, although the finding that this ASV was absent in 2/13 samples confirmed to contain RBS-As via microscopy challenges this hypothesis. One possibility is that the RBS-As were present in sufficiently low relative abundance in those two samples as to not be detected, despite deep amplicon sequencing. Another possibility is that RBS-As are not this *Alcaligenaceae* taxon,

and rather the *Alcaligenaceae* taxon is a ubiquitous member of the dolphin oral microbiota and therefore showed up frequently in our sequencing-based analyses. In such a case, it would suggest that either the RBS-As are class Betaproteobacteria that did not lyse in the sequencing-based experiments or that the FISH results were a false positive, despite the stringent and controlled conditions under which the experiment was performed.

Obtaining a species-level identification for RBSs via sequencing-based methods will be extremely challenging for numerous reasons, such as the frequent close proximity of RBSs with other small cells that were likely mixed with RBSs during micromanipulation or recalcitrance to laboratory lysis. Importantly, we are hesitant to exclude candidate identities based on not being present in the mini-metagenomics experiment in all positive RBS samples, since technical limitations could have resulted in false negatives. For example, a thick cell wall or obstruction preventing reagents from reaching the RBS by the micropipette needle could have interfered with lysis of the cell membrane and impeded DNA extraction. Conversely, a positive result in a negative control may have arisen due to non-specific read mapping, contamination of genome bins with material from true contaminants, or cell-free DNA. Culturing-based approaches may ultimately be the most promising route forward for identifying RBSs, although many combinations of parameters will likely need to be tested to find satisfactory conditions for RBS growth. Regardless of the taxonomic identity of the RBSs, the novel structural features that have evolved in these microorganisms are intriguing and highlight the discovery potential for further study.

The paired nature of segments in RBSs can likely be ascribed to their longitudinal mode of binary fission, as seen in the family *Neisseriaceae* genera *Alysiella*, *Simonsiella*, and *Conchiformibius*, as well as *S. poulsonii, Ca.* T. oneisti, and *Ca.* T. hypermnestra[31,32,35,54]. In the family *Neisseriaceae* member *S. muelleri*, sheets are thought to help cells remain physically anchored in the oral cavity when rapidly shedding epithelial cells slough off[34]. We hypothesize that the same may be true for RBSs, which inhabit a similar environment and whose morphology may have undergone convergent evolution due to similar evolutionary pressures. Longitudinal binary fission may be an even more general characteristic that is selected in response to the need to form a secure attachment to a substrate. The segments at the ends of RBSs are often shorter than those closer to the center, suggesting that there may be a mechanism by which the growth of segments is determined by their spatial positioning within an RBS. The RBSs present another case example for future evolutionary studies focused on understanding the drivers and genetic basis of bacterial multicellularity and longitudinal division.

CryoTEM images suggested that RBS-As are encapsulated by a periodic surface covering, which may be an S-layer or a new crystalline structure. S-layers are self-assembling, crystalline arrays of single proteins or glycoproteins that coat the exterior of some bacteria and archaea[55,56]. While their exact function varies widely across microorganisms and is often unknown, S-layers are hypothesized to confer beneficial functions given their high metabolic cost (the S-layer comprises up to ~20% of the total protein synthesized by cells), their ubiquity across microbes, and their multiple evolutionary origins[55,56]. If segments in RBS-As correspond to individual cells, the production of the periodic surface covering may represent cooperation between cells within RBS-As. Cooperative synthesis of a single, shared periodic surface covering by multiple cells could have evolved since close kin (other cells in an RBS-A) have limited dispersal ability and are therefore situated in close physical proximity. RBS-As would benefit from the cooperative production of a single periodic surface covering around a population of cells rather than around each individual segment by reducing the surface area required to cover all cells, and such an advantage could even have contributed to selection for aggregation. An additional and not mutually exclusive possibility is that the periodic surface covering may help to maintain the ultrastructure of segments within an RBS-A, similar to archaea such as *Thermoproteus tenax*[57].

One of the most striking features of RBS-As is their pilus-like appendages. At present, there are five characterized classes of pili in Gram-negative bacteria (chaperone-usher, curli fibers, F-type, type IV, and type V) and two general types of pili in Gram-positive bacteria (short, thin rods and longer, flexible, hair-like filaments)[45,46]; other pilus-like appendages have been documented, such as hami in archaea[21]. To the best of our knowledge, characterized bacterial pili all consist of single appendages that exist as independent units. By contrast, the pilus-like appendages that protrude from RBS-A segments exhibit an unusual architecture involving heterogeneous bundles of filaments that often splay out at the tips. These observations raise the question of whether the RBS-A appendages represent a novel type of assembly of pilin subunits or are a completely distinct class of appendages.

Extensive investigation of hundreds of cryoEM images and dozens of cryoET tomograms enabled visualization of the structure of many RBS-A features at close to nanometer resolution. Future studies of RBSs may benefit from imaging with phase-plate optics that dramatically increase image contrast[58] following specimen preparation methods that thin cells into lamellae by FIB milling coupled with SEM at cryogenic temperatures[48,49]. Our data suggest that cryoCLEM will be needed to enable the production of thin lamellae for RBS-As since the RBS-A cell body seems to be thicker than the limit allowed for cryoET experiments, and yet too thin for RBS-As to be readily found by cryoSEM prior to cryoFIB milling without fluorescent labels. Successful cryoCLEM+cryoFIB-SEM experiments followed by cryoET could enable more comprehensive analyses of the community of RBSs and their cell body beyond the thin periphery as well as visualization of subcellular components of interest at higher resolution via subtomogram averaging.

The vast majority of microorganisms on Earth lack isolated representatives[19]. Sequencing-based analyses have proved invaluable in exploring and describing said diversity, yet cannot be used to explore all aspects of the biology of microorganisms. Notable blind spots in our understanding of uncultured organisms include the unique genes and corresponding structural and functional features that have evolved within these lineages. While the use of advanced imaging techniques to visualize microbes can provide insight into the biology of uncultured lineages, a shift toward a more multifaceted approach drawing on many disciplines and techniques will be required to create a comprehensive view of this biological dark matter[59,60].

## Methods
To maximize reproducibility, a list of the reagents and resources used in this study, as well as their source and identifier, is provided in Supplementary Table 5.

### Experimental model and subject details
Oral swab samples were obtained from bottlenose dolphins (*Tursiops truncatus*) managed by the U.S. Navy MMP Biosciences Division, Space and Naval Warfare Systems Center Pacific, San Diego, USA. The earliest sample containing RBSs was collected on April 1, 2012, and the latest on March 24, 2022. Swab samples were obtained using sterile foam Catch-All sample collection swabs (Epicenter, WI, Cat. #QEC091H). Samples collected in 2012 were obtained by swabbing the left gingival sulcus. Samples collected in 2018 were obtained by swabbing the palate, the tongue, and the left gingival sulcus (all three surfaces for each swab). Samples collected in 2022 were obtained from the palate, buccal surface, or left gingival sulcus. Of the 2022 samples of the gingival sulcus, 5 were stored in 20% glycerol. All other swab samples were dry frozen. The swabbing protocol adhered to the guidelines described in the CRC Handbook of Marine Mammal Medicine.

The MMP is accredited by the Association for Assessment and Accreditation of Laboratory Animal Care International and adheres to the national standards of the United States Public Health Service Policy on the Humane Care and Use of Laboratory Animals and the Animal Welfare Act. As required by the U.S. Department of Defense, the MMP's animal care and use program is routinely reviewed by an Institutional Animal Care and Use Committee (IACUC) and by the U.S. Navy Bureau of Medicine and Surgery. The animal use and care protocol for MMP dolphins in support of this study was approved by the MMP's IACUC and the Navy's Bureau of Medicine and Surgery (IACUC #92-2010, BUMED NRD-681).

## Microscopy sample preparation

To separate cells from swabs, swabs were immersed in 1X PBS (~50–100 μL, depending on cell density) in microcentrifuge tubes. Tubes were vortexed vigorously for ~10 s and lightly centrifuged to remove liquid from tube caps. The resulting solution was used for microscopy.

## Light microscopy

Approximately 1 μL of cell solution in PBS was spotted onto an agarose pad (1% agarose in PBS) and imaged with an Eclipse Ti-E inverted microscope with a 100X (NA: 1.4) objective (Nikon, Tokyo, Japan). To determine DNA localization, cells were stained with DAPI at a final concentration of 0.5 μg mL$^{-1}$ for 5 min prior to imaging using emission/excitation spectra of 340/488 nm. High-throughput, automated imaging of dolphin oral samples via phase contrast microscopy was achieved using the Strain Library Imaging Protocol[38] to capture 226 fields of view for each of the samples collected in 2018, and 100 fields of view for those collected in 2022.

## Gram and FM4-64 staining

Gram staining was performed using a Gram Staining Kit (Sigma Aldrich, cat. #77730-1KT-F) following the manufacturer's protocol. Cells were imaged using a bright field microscope with a 100X objective (Nikon). FM4-64 dye (ThermoFisher Scientific, cat. #T13320) was applied directly to dolphin oral swab samples following the manufacturer's protocol. The FM4-64 dye did not stain any part of the RBSs.

## RBS-A cryofixation and cryoEM/ET data acquisition

A solution of cells in PBS (2.5 μL) was applied to glow-discharged 200-mesh copper, holey-carbon Quantifoil grids (Quantifoil, Großlöbichau, Germany, Cat. #Q2100CR1) or gold GridFinder Quantifoil grids (Quantifoil, Großlöbichau, Germany, Cat. #LFH2100AR2), followed by application of 2 μL of 15 nm gold fiducial solution to both sides of each grid. Grids were blotted for 5 s and plunge-frozen in liquid ethane cooled by liquid nitrogen to approximately −195 °C using an EM GP Plunge Freezer (Leica, Wetzlar, Germany).

Samples were loaded into one of two microscopes: a Titan Krios G3 operated at 300 kV with an energy filter (20-eV slit width), or a Titan Krios G4 operated at 300 kV without an energy filter. Both microscopes were equipped with a K2 Summit direct electron detection device (Gatan, Pleasanton, USA) used to record micrographs. Data were acquired semi-automatically in counting mode using SerialEM (v. 3.8)[61]. CryoEM/ET imaging parameters are provided in Supplementary Table 6.

## CryoEM/ET data processing

Montages were blended and binned 4-fold or greater using the IMOD v. 4.12.9 "blendmont" algorithm[62] and normalized, band-pass filtered, rotated, and cropped for display purposes using EMAN2 v. 2.39[63]. Fifteen out of sixteen tilt series were suitable for tomographic reconstruction in IMOD v. 4.12.9. Tilt series with sampling at 7.5 Å pixel$^{-1}$ were down-sampled by 2-fold and those with sampling at 3.48 or 3.75 Å pixel$^{-1}$ were down-sampled by 4-fold. Images with artifacts such as

excessive charging, drifting, large ice contamination creeping in at high tilts, or excessive thickness at high tilt were excluded from 12 of the tilt series prior to manual gold-fiducial-based alignment; up to 13 images were removed out of the 41 images in the original raw tilt series.

Tomograms were reconstructed using standard weighted back-projection and a SIRT-like filter (mimicking 16 iterations) and were band-pass-filtered and further binned by 2-fold in most cases for feature annotation, segmentation, movie production, and other display purposes. Tomogram thickness was estimated by visually identifying the smallest and largest z-slices with visible RBS-A or ice contamination densities and converting the number of slices to nanometers. Subtomogram averaging was attempted using EMAN2 v. 2.39[64,65] for globular densities suspected to be ribosomes, matrix densities under the outer membrane, patches of the periodic surface covering, and regions of pilus-like appendages, but no interpretable structures with resolution better than ~50 Å were obtained. The ranges of thickness and length for the pilus-like appendages were derived by visually scanning the slices in the tomograms for the thinnest individual filaments and thickest bundles perceptible to the naked eye and measuring their dimensions in binned-by-4 tomographic slices using the measuring tape tool of EMAN2 e2display.py. The repeat distance of the periodic surface covering was measured manually in a similar fashion as the pilus-like appendages from tomographic slices, with ~10–20 measurements from each of the three tomograms displaying at least small regions where the repeat was discernible. This quantification yielded a range between ~6 and 10 nm, suggesting that either the layer components are flexible or that the underlying structure can yield different apparent distances between its subunits depending on the angle at which it is sliced. In addition, regions showing the pattern much more clearly in higher-magnification montage two-dimensional projection images were cropped out, rotated to lie in a horizontal plane, filtered, and masked to compute line-density profiles parallel to the outer membrane.

We initially carried out tomographic annotation of three features (periodic surface covering, lipid membranes, and pilus-like appendages) for three tomograms using EMAN2's semi-automated two-dimensional neural network-based pipeline[66] and performed manual clean-up of false positives in UCSF Chimera v. 1.16[67]. The output annotation probability maps from EMAN2 v. 2.39 were turned into segmentations by applying a visually determined threshold and multiplying the contrast-reversed tomograms by the thresholded annotation map. The segmentations were low-pass-filtered with EMAN2 v. 2.39 to smooth out noise. However, since the complexity of subcellular structures was not captured by the semi-automated annotations, we applied a similar process to generate segmentations of five features (pilus-like appendages, periodic surface covering, outer membrane, matrix, and inner membranes) using manual annotations performed with IMOD v. 4.12.9, following a recent protocol that increases manual annotation efficiency[68]. Snapshots for Fig. 6 displaying RBS-A features in color as well as Supplementary Movies 2 and 3 showing segmentation results were produced with UCSF Chimera v. 1.16.

## CryoFIB-SEM

We first identified a grid that contained RBS-As using light microscopy. Using an Aquilous CryoFIB (ThermoFisher Scientific, MA, USA), we created five lamellae using 30 kV, 30 pA current and 5 μs duration time. The samples were then loaded into a cryoTEM for sample observation and data collection. We were unable to identify RBSs in the lamellae.

## Fluorescence in situ hybridization

Cell cultures of axenic *Escherichia coli* MG1655, non-axenic *Skeletonema costatum* LB 2308 (UTEX Culture Collection of Algae at the University of Texas at Austin, Austin, TX, USA), *Caulobacter crescentus* CB15N, and *Simonsiella muelleri* ATCC 29453 were prepared as controls. *E. coli* was cultured in LB broth and grown at 37 °C, *S. costatum*

was cultured in Erdschreiber's Medium at 20 °C with a ~12 h light and ~12 h dark cycle, *C. crescentus* was cultured in PYE medium at 30 °C, and *S. muelleri* was cultured in BSTSY medium (2.75% (w/v) Tryptic Soy Broth, 0.4% (w/v) yeast extract, 10% bovine serum) at 37 °C.

All FISH probes were ordered from Integrated DNA Technologies (Coralville, USA) with HPLC purification. Probe sequences and fluorescence labels are as follows: Euk-1209: 5′-GGGCATCACAGACCTG-/3Alx660/−3′, Bact338: 5′-GCTGCCTCCCGTAGGAGT-/Alx488/−3′, BET42a: 5′-/Alx594/-GCCTTCCCACTTCGTTT-3′, GAM42a: 5′-/Alx488/-GCCTTCCCACATCGTTT-3′, nonEUB: 5′-/Cy5/-ACTCCTACGGGAGG-CAGC-3′.

Cells from controls and RBS-As were collected in microcentrifuge tubes. To ensure sufficient biomass from dolphin oral swabs, cells from four swabs were condensed into a single tube. The FISH protocol was adapted from ref. 69. Cells were fixed in 1 mL of 3.7% formaldehyde solution (800 µL of DEPC-treated water, 100 µL of 10X PBS, 100 µL of 37% formaldehyde) for 30 min with gentle shaking at 700 rpm. Cells were then washed twice in 1 mL of 1X PBS, and permeabilized in a mixture of 300 µL of DEPC-treated water and 700 µL of 200-proof ethanol with gentle shaking at 700 rpm for 2 h. Probes were added to 50 µL of hybridization solution to a final concentration of 1 µM per probe set. For BET42a and GAM42a, the hybridization solution contained 55% formamide solution (4 mL of DEPC-water, 1 g of dextran sulfate, 4.85 mL of formamide, 1 mL of 2X SSC, brought to a total volume of 10 mL with DEPC-treated water); for other probes, the hybridization solution contained 40% formamide (5 mL of DEPC-water, 1 g of dextran sulfate, 3.53 mL of formamide, 1 mL of 2X SSC, brought to a total volume of 10 mL with DEPC-treated water). For BET42a and GAM42a, cells were incubated in 50 µL of hybridization buffer with probes at 46 °C for 1 h; for other probes, cells were incubated overnight in 50 µL of hybridization buffer with FISH probes at 30 °C. Cells were washed twice using a wash solution (2 mL of 20X SSC buffer, 7.06 mL of formamide, 10.94 mL of DEPC-treated water) and resuspended in 2X SSC buffer. One microliter of cells was mounted onto 1% agarose pads containing PBS and 5 µg mL⁻¹ of DAPI for imaging. Imaging data were processed using FIJI v. 2.0.0[70].

### 16S rRNA amplicon sequencing and processing

Fifty-four dolphin oral samples were selected for 16S rRNA gene amplicon sequencing. Genomic DNA was extracted from dolphin oral samples and 32 negative (PBS) controls using the DNeasy UltraClean 96 Microbial Kit (Qiagen Cat. #10196-4) following manufacturer's instructions. The 16S rRNA V4 region was amplified using 515F and 806rB primers using Platinum™ II HotStart PCR Master Mix (Thermo-Fisher Cat. #14000013). The PCR products were pooled at equal volume and gel-purified. Final purification was performed using Macherey-Nagel NucleoSpin Gel and PCR Clean-up, Mini Kit (Fisher, Cat. #740609). Amplicons were sequenced on the Illumina MiSeq platform with 250-bp paired reads at the Stanford Chan Zuckerberg Biohub Facility, resulting in a median read depth of 92,077 reads (min: 50,772, max: 265,108).

Demultiplexing was performed using Bcl2Fastq v. 2 (Illumina, CA, USA). ASVs were inferred using DADA2[71] v. 1.16.0, following guidelines in the "Big Data Workflow" (https://benjjneb.github.io/dada2/bigdata_paired.html). Taxonomic affiliations were assigned using the SILVA 138 SSU database[72] as a reference. Forward and reverse reads were trimmed to 240 and 180 nt, respectively. This pipeline yielded a total of 1116 taxa and 5,339,751 reads across the 54 samples. ASVs were analyzed using phyloseq v. 1.28.0[72].

### Spike-in experiment

To confirm that our DNA extraction, amplification, sequencing, and bioinformatic analysis pipeline was able to identify members of the family *Neisseriaceae*, we performed a spike-in experiment. We first selected a dolphin oral sample that contained RBS-As, as

determined by phase-contrast microscopy. We then created two aliquots of this sample. To the first aliquot, we spiked in cells from an *S. muelleri* pure culture at a 1:1 ratio. The other aliquot was left untouched. These two samples, along with an aliquot of the *S. muelleri* pure culture (positive control), underwent DNA extraction, PCR amplification, and sequencing protocol as described above. The three samples were sequenced in a single Illumina MiSeq run, which was not in the same lane as the other amplicon samples from this study. Note that by PCR amplifying *S. muelleri* in the same lab environment as the negative control sample and then sequencing it on the same lane, some cross-contamination is to be expected. The first time the negative control sample was sequenced (in the absence of *S. muelleri* pure cultures in the laboratory), no family *Neisseriaceae* amplicons were detected.

### Mini-metagenomics

To obtain candidate identities for RBS-As, we employed a mini-metagenomics approach. To limit contamination by foreign DNA, reagents, tubes, and PBS were treated with 11.4 J cm⁻² of ultraviolet light following the guidelines in ref. 72. RBS-As were visualized using an Olympus IX70 inverted microscope (Olympus, Waltham, USA) with a 40X objective and Hoffman modulation optics. An Eppendorf TransferMan micromanipulator (Eppendorf, Hamburg, Germany, Cat. #5193000020) with a SAS-10 microinjector was used to capture RBS-As with Polar Body Biopsy Micropipettes (30° angled, beveled, and polished with an inner diameter of 13–15 µm) (Cooper Surgical, Målov, Denmark, Cat. #MPB-BP-30). After an RBS-A or chain of RBS-As was acquired, the micropipette tip was transferred to a collection tube containing 1X PBS and crushed into the tube to ensure the RBS-A(s) was deposited in the tube; this precaution was adopted because RBS-As frequently stuck to the glass micropipette and could not be dislodged. No dolphin cells were captured, although cell-free DNA and small, non-target cells from the sample were likely acquired as contaminants along with RBS-As based on the propensity of the latter to attach to other species (Fig. 5e). Four tubes of RBS-As were collected (sample names RBS1-4), along with four negative-control tubes (NEG1-4). Negative controls consisted of draws of PBS from the same sample that did not contain any visible cells and were otherwise treated identically to RBS-A-containing samples.

DNA from each tube was amplified via MDA using the Repli-g single-cell kit (Qiagen, Hilden, Germany, Cat. #150343) according to the manufacturer's protocol. DNA was purified using a Zymo Clean and Concentrate Spin Column (Zymo Research Corporation, Irvine, USA, Cat. #D4013) and libraries were prepared using the Kapa Hyper Prep Kit (Kapa Biosystems, Wilmington, USA, Cat. #KK8504) at the W.M. Keck Center for Comparative Functional Genomics at the University of Illinois, Urbana-Champaign. The eight libraries were sequenced using the Illumina MiSeq 2 × 250 nt P2 V2 platform. RBS-A samples RBS1, RBS2, NEG1, and NEG2 were pooled and sequenced across a single lane that produced 11,371,243 read pairs, and samples RBS3, RBS4, NEG3, and NEG4 were pooled and sequenced across 1.5 lanes, collectively producing 19,615,690 read pairs. Sequencing adapters were computationally removed at the Keck Center.

Reads from all eight samples were co-assembled using SPAdes v. 3.11.1[73] with the single-cell (−sc) and careful (−careful) modes specified. A total of 61,973,866 read pairs were used for assembly, resulting in 1406 scaffolds ≥5 kbp long with a total length of 17,438,233 bp and an N50 of 14,592 for scaffolds ≥5 kbp long. Protein-coding genes were identified using Prodigal v. 2.6.2[74]. Per scaffold average coverage was calculated by mapping reads per sample against the co-assembly using bowtie2 v. 2.2.4[75], using the samtools v. 1.6 depth function[76] to calculate per-base read coverage, and a custom script to calculate average per-base read coverage per scaffold.

A search for the *amiC2* gene in bins recovered from the mini-metagenomics experiment was performed by querying Pfam

alignment PF01520 against each genome's protein sequences, using HMMER suite v. 3.1b2[77].

To determine the taxonomic identity of sequenced cells, we employed a genome-resolved approach. Assignment of scaffolds to genome bins was performed using the tetranucleotide frequencies of all scaffolds ≥5 kbp long over windows of 5 kbp, as described in ref. 78. Results were computed and visualized using the Databionics ESOM Tools software v. 1.1[79], leading to the reconstruction of 18 genome bins (Supplementary Fig. 3). To refine bins, we removed scaffolds for which <50% of keys were assigned to the bin. Scaffolds <5 kbp long were not binned. The completeness and contamination per bin were assessed using CheckM v. 1.0.7[80]. To evaluate how representative binning was of the genomes that were sequenced, we estimated the number of prokaryotic genomes expected to be recovered by searching the metagenome assembly for a set of 16 bacterial single-copy genes (bSCGs) assumed to be present in every genome in a single copy[81], namely ribosomal proteins L2, L3, L4, L5, L6, L14, L15, L16, L18, L22, L24, S3, S8, S10, S17, and S19. Alignments for these proteins (PF00181, PF00297, PF00573, PF00281, PF00347, PF00238, PF00828, PF00252, PF00861, PF00237, PF17136, PF00189, PF00410, PF00338, PF00366, and PF00203) were obtained from the Pfam database[82] (accessed March 2019) and queried against our dataset using HMMER suite v. 3.1b2[77]. The median number of each bSCG was 10, suggesting ~10 prokaryotic genomes were represented in our sequencing dataset. In the case of the family *Alcaligenaceae* genome of interest, the 16S rRNA gene was manually extended from the end of a scaffold; using bowtie2 v. 2.2.4 we ensured that the reads supported the final sequence.

Taxonomic identification of bins posed a challenge since 16S/18S rRNA genes were not reliably amplified/sequenced/assembled, and genomes were partial with few phylogenetically informative bSCGs present in the dataset. Hence, we used BLAST v. 2.2.30[83] to query all protein-coding genes from each genome against the NCBI non-redundant protein database using an e-value of $10^{-10}$ and taxonomic assignments were made based on the closest protein match. Genome bin taxonomic assignments were considered highly likely if ≥50% of the top BLAST hits originated from a single taxon and were considered plausible if <50% but ≥33% of the top BLAST hits originated from a single taxon.

There are numerous approaches by which one could assess whether a bin is "present" in a sample, each with largely arbitrary thresholds. We focused on the relative abundance of each bin per sample (Supplementary Table 4) as it accounts for the length of each bin and allows for comparisons between samples with different numbers of read pairs[16].

### Confirmation of the taxonomic identity of bin 16

Maximum likelihood phylogenies of the family *Alcaligenaceae* were inferred using the 16S rRNA gene (Supplementary Fig. 4a) and ribosomal protein S3 (Supplementary Fig. 4b) to confirm the taxonomic affiliation of bin 16, which was recovered from the mini-metagenomics experiment. Gene/protein sequences were acquired for each characterized genus in the family *Alcaligenaceae*, as shown on the NCBI Taxonomy Browser (accessed August 2022), when such sequences were available in the NCBI system (some genera have scant or no genomic representation). We additionally performed BLAST[83] v. 2.2.30 queries of the bin 16 16S rRNA gene and rpS3 protein against the nr/nt and nr databases (accessed August 2022), respectively, and included the top 10 most similar sequences. 16S rRNA gene sequences were aligned using SINA[84] v. 1.2.11, using the SILVA SSU database release 138.1 as a reference, and columns containing >3% gaps or rows with <50% sequence were removed. rpS3 protein sequences were aligned using Clustal Omega[85,86] v. 1.2.4 and columns containing >5% gaps or rows with <50% sequence were removed. Both phylogenies were inferred using PhyML[87] v. 3.1 with 1000 bootstrap replicates, with model selection performed using smart model selection[88] (GTR+R for

the 16S rRNA gene and Q.yeast+G+I for the rpS3 protein). Trees were visualized using iTOL[89] v. 6.

### Attempt at culturing RBSs

Four dolphin oral samples confirmed to contain RBSs were selected for culturing efforts. For each sample, one milliliter of sterile PBS was added to a 1.5-mL Eppendorf tube containing the oral swab sample. For liquid culturing, 600 μL of each sample were used to inoculate 3 mL of BSTSY[90] (2.75% (w/v) Tryptic Soy Broth, 0.4% (w/v) yeast extract, 10% bovine serum), SHI[91], or mSHI media (SHI supplemented with 0.9 g/L NaCl, 2.5 g/L $K_2PO_4$, 0.84 g/L $NaHCO_3$, 0.17 g/L $CaCl_2$, 0.04 g/L $MgCl_2 \cdot 6H_2O$, and 5 g/L dextrose). BSTSY was selected for its use in successfully culturing bacteria (*S. muelleri* specifically) from the oral cavities of various mammals[90]. SHI was selected because this medium was designed to sustain high-diversity communities derived from the human oral microflora. mSHI (modified SHI) was included as a higher-salinity version of SHI in an attempt to further mimic the conditions that might be found in the oral cavity of dolphins. Inoculation was repeated under anaerobic conditions in an anaerobic chamber (COY Lab Products, Grass Lake, USA); note that all samples were unavoidably exposed to atmospheric oxygen prior to culturing. Cultures were incubated at 37 °C to mimic the body temperature of dolphins. No RBSs were detected in liquid media by visual screening under a microscope after ~24, ~48, ~72, and ~96 h. For solid-surface culturing, ~$10^3$–$10^4$ cells (verified by microscopy to contain RBS-As) were directly plated onto BSTSY or BHI-blood (BHI medium supplemented with 5% sheep blood) agar plates, and incubated at 37 °C with or without oxygen, respectively. No colonies grew on the BSTSY plates after 3 weeks of incubation; the colonies grown on the BHI-blood plates were screened using microscopy and no RBSs were visible. By contrast, a control of *S. muelleri* streaked onto BSTSY plates developed visible colonies after 1–2 days of incubation with oxygen at 37 °C, and the colonies were verified under microscopy to consist of cells with the morphology expected of *S. muelleri*.

### Statistics and reproducibility

Given the exploratory nature of this descriptive study, most experiments to characterize the RBSs were performed a single time.

### Reporting summary

Further information on research design is available in the Nature Portfolio Reporting Summary linked to this article.

## Data availability

Sequencing data for this project are available through NCBI BioProject PRJNA174530. Raw reads for the amplicon survey were deposited to SRA and are associated with BioSamples SAMN32739817-69 and SAMN19012476. Data for the spike-in experiment are associated with BioSamples SAMN32869723-5. Raw reads for the single-cell genomics experiment were also deposited to SRA; the captured RBSs and negative controls were physically derived from a single oral swab represented by BioSample SAMN19012476, while the reads from the eight experimental replicates (four RBSs, four negative controls) are each individually associated with BioSamples SAMN19022663-SAMN19022670. The co-assembly of scaffolds ≥5 kb in length from the single-cell genomics experiment was deposited as a Whole Genome Shotgun project at DDBJ/ENA/GenBank under the accession JAHCSF000000000, following the removal of human-derived sequences. The version described in this paper is JAHCSF010000000. Genome bins 2–5 and 7–18 from the single-cell genomics experiment have been deposited as a Whole Genome Shotgun project at DDBJ/ENA/GenBank under accessions JAGYHI000000000-JAGYHX000000000. The versions described in this paper are JAGYHI010000000-JAGYHX010000000. Genome bin 1 (human) was not deposited. Scaffolds for genome bin6 (<100,000

nucleotides) were deposited as a non-genome GenBank submission under accession numbers MZ126582-MZ126593. Public datasets and databases used in this study are as follows: the SILVA SSU database release 138.1 was used as a reference for assigning taxonomic identities to ASVs. The NCBI nr/nt, nr, and taxonomy databases (accessed August 2022) were used to obtain 16S rRNA gene sequences ($n = 77$) and ribosomal protein S3 sequences ($n = 63$) for representatives of genera of the family *Alcaligenaceae*. The NCBI accession numbers for these sequences are available in Supplementary Figs. 4 and 5, respectively. Finally, we performed analyses with alignments sourced from the Pfam[82] database. Pfam alignment PF01520 was used to search genome bins for AmiC2 proteins (accessed August 2022). The following Pfam alignments were used to search for 16 bacterial single-copy genes (accessed March 2019): PF00181, PF00297, PF00573, PF00281, PF00347, PF00238, PF00828, PF00252, PF00861, PF00237, PF17136, PF00189, PF00410, PF00338, PF00366, and PF00203.

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

## Acknowledgements

We thank members of the Relman lab for insightful discussions, Katharine Ng and Brian Yu for assistance in attempts to capture RBS-As using laser capture microdissection and microfluidics, respectively, Melissa Clark for assistance with Gram staining, Michael Schmid for analysis of the periodic surface covering on RBS-As, and especially Celeste Parry and colleagues at the U.S. Navy Marine Mammal Program in San Diego, CA for collecting dolphin oral samples. This work was supported by NIH grant P41GM103832 (W.C.), a James S. McDonnell Postdoctoral Fellowship (H.S.), the Austrian Science Fund (T.V., S.B.), and the Thomas C. and Joan M. Merigan Endowment at Stanford University (D.A.R.). K.C.H. and D.A.R. are Chan Zuckerberg Biohub Investigators.

## Author contributions

Conceptualization: N.K.D., K.C.H., D.A.R. Methodology and investigation: N.K.D., J.G.G.-M., H.S., M.M., C.D., A.I.C., G.-H.W., T.V., S.B., B.B., K.C.H., W.C., D.A.R. Formal analysis: N.K.D., J.G.G.-M., H.S., C.D. Visualization: N.K.D., J.G.G.-M., H.S., C.D. Writing—original draft: the manuscript was mainly written by N.K.D. and J.G.G.-M. based on N.K.D.'s original draft, with revisions by all other authors. Writing—review and editing: all authors. Supervision: K.C.H., W.C., D.A.R.

## Competing interests

The authors declare no competing interests.
