## [Peer Review File · Nature Communications]

Previously uncharacterized rectangular bacterial structures in the dolphin mouthReviewer #1 (Remarks to the Author):

This study describes an uncultivated microbe with unusual rectangular morphology examined with different imaging approaches. This examination of RCUs revealed distinct organization of DNA and membranes within the rectangularly shaped structure, suggesting RCUs were likely multicellular structures whose members divide along a transverse axis. CryoEM and cryoET revealed oddly shaped structures similar to pili. Attempts were also made to identify the phylogenetic identity of the microbe and sequencing its genome, but clearly linking a phylogenetic assignment to the RCUs was not possible.

This is an entirely descriptive paper, which is ok when describing such a morphologically interesting system, and there is a large amount of work that went into imaging and attempting to isolate and/or sequence the RCUs. I do not have any serious concerns about the data and methods presented. Unfortunately, the description seems incomplete, and we only have a partial story. It's not clear if RCUs are unicellular polyploid cells or multicellular structures, and the manuscript language seems to alternate referring individual RCUs as a 'cells' rather than aggregates, etc. The data point towards the latter, but this could be more thoroughly addressed with additional staining and imaging, e.g. with a membrane stain like FM 1-43 and confocal microscopy. Or perhaps this can be addressed with a better explanation and illustration of the gram staining and cryoEM to more clearly show the RCUs are composed of clearly separate cells.

In addition, RCUs ultimately were not assigned to a phylogenetic group or genome. I realize the difficulty, but more could be done. For example, the authors identify a limited number of likely genomes that belong to Bacteroidetes, gammaproteobacterial, and epsilonproteobacteria groups. FISH worked well with Eub338 probes and perhaps it will also work with probes for these groups, or possibly with probes at a finer phylogenetic scale without the authors needing to design and validate new probes. Finally, there was no connection between genomes and structures. For example, did any MAGs or unassembled reads have any similarity to genes involved in pili formation or similar structural elements?

I think more definitely establishing any of these points would probably be enough to complete the story and make it a publishable unit, but I'm afraid that that manuscript does not meet that threshold yet in my opinion

Reviewer #2 (Remarks to the Author):

Because bacterial cell biology has been focusing on model bacteria there is a tremendous need of investigations such as this one conducted by Dudek et al. In spite not being able to isolate them in pure culture, the authors discovered that:

- 1) multicellular, likely longitudinally dividing bacteria (of unknown phylogenetic placement) may inhabit the dolphin mouth**
- 2) each multicellular unit of longitudinally dividing bacteria contains multiple stripes of DNA that can be shorter or longer (not quantified)**
- 3) they have pili-like structures that are splayed out.**

The most remarkable result of this study, in my opinion, is the discovery of pilus-like appendages which protrude as bundles of threads partially parallel to each other but splayed-out at the tips (number 3). This is, to the best of my knowledge, unprecedented.

There are however a number of claims/conclusions which are not supported:

- 1) Already in the title and throughout the manuscript the authors claim to have**

characterised "rectangular" bacteria: although the multicellular units might appear rectangular, what are the evidence(s) that each individual bacterium is rectangular, i.e. each cell has 90 degrees, non-rounded corners? When I look at the beautifully cryoEM-based 3D reconstructions, the individual cells look rod-shaped to me. If, with "rectangular", the authors refer to the filament, then, perhaps, Rectangular Multicellular Unit (RMU) would be more appropriate than Rectangular Cell-like Unit (RCU).

2) The authors claim that the "rectangular" units are not belonging to the genus *Simonsiella* because this genus was not detected in all their sequencing efforts: was the sequencing depth sufficient to detect *Simonsiella* even if it had been present in low abundance? The authors need to show rarefaction curves to support their claim/to make sure that the sequencing was deep enough to discover rare taxa. Alternatively, failure in amplifying *Simonsiella* 16S rDNA by qPCR or PCR with *Simonsiella*-specific primers is needed to support the claim that "the rectangular units are distinct from *Simonsiella*". Another possibility is to perform FISH with *Simonsiella* (or *Neisseriaceae*) 16S rRNA gene-specific probe(s).

Additional comments:

line 29: what do the authors mean by "microbial structure"? Please clarify

line 30, 134: what is rectangular: the cell or the filament? If the authors are referring to the individuals cells, these/their corners should be measured

line 31: please specify that the genomic DNA underwent an amplification step.

line 33: may be distinct from *Simonsiella* (see my major concerns above).

line 36: "membrane-bound segments": please clarify.

line 45: the authors should provide evidence for the existence of an S layer.

line 103: not only *Candidatus T. hypermenestrae*, but also the fruit-fly endosymbiont *Spiroplasma poulsonii* (Ramond et al., 2016) and *Candidatus T. oneisti* divide longitudinally (Leisch et al., 2012)

line 106: up to now, only a species of *Simonsiella* is known, *S. muelleri*.

line 108: *Simonsiella muelleri* is cultivable (see for example Hedlund BP, Tønjum T. *Simonsiella*. *Bergey's Manual Systematics of Archaea and Bacteria* [Internet]. 2015;1–12).

line 134: multicellular units, not individual cells, contain multiple bands

line 139: we understand that they are prevalent in dolphin mouths (they were found in 20 out of 28) but can the author provide an estimate of their relative abundance?

line 141: do the authors mean a stack of squared pancakes viewed from the side?

line 148: "the individual DAPI-stained bands tended to be longer": how much longer? Can the authors please specify?

line 152: representative image (D,E) shows an RCU that is longer than the RCUs in (B,C). Or do the authors refer to the single cells?

line 153, line 156: can the authors clarify/estimate the occurrence of each morphotype?

line 163: neither *Candidatus T. hypermenestrae* cells nor *Simonsiella* cells/filaments,

based on the literature, are rectangular (Hedlund BP, Tønjum T. *Simonsiella*. *Bergey's Man Syst Archaea Bact* [Internet]. 2015;1–12). They all are considered rod-shaped.

line 169: *Simonsiella muelleri* or *Simonsiella* sp.?

line 170: the fact that it worked on sea lion samples is no guarantee that it works on dolphin samples

line 197: the authors did not employ single-cell genomics. Single-cell genomics refers to the process of separating communities into single units, the amplification of DNA of single cells and its subsequent sequencing. Instead, the authors micromanipulated the communities via suction and, most likely, amplified DNA of multiple cells (the "negative" samples reflect this).

line 198: In principle yes, but, in fact, the phi29 polymerase - commonly used for amplification - has a strong bias towards sequence composition, i.e. GC-rich stretches are usually not well amplified, and hence, are underrepresented in sequencing.

lines 211-212: to be sure that the rectangular units are neither diatoms nor *Simonsiella* the authors should show that diatoms and *Simonsiella* cells can be successfully lysed and their DNA amplified by phi29.

line 281-282: "such as the high and variable curvature of the regions with a continuous periodic surface covering": please clarify.

line 304: in my opinion, the wording "single-cell" genomics is not appropriate (see above)

line 320: on the basis of what do the authors think that the rectangular units, which thrive in the mouth of dolphins, an aerobic habitat, could be strictly anaerobe?

line 327: *S. muelleri*, *Spiroplasma poulsonii*, *Ca. Toneisti* and *Ca. T. hypermnestrae*

line 544: did the authors do negative controls?

line 547: concentration of reagents and cycling conditions are missing

line 554: how many reads per samples?

line 620: how do the authors searched the metagenome assembly?

line 924: on the basis of what do the authors claim that the black dots correspond to storage granules or lipid droplets?

Reviewer #3 (Remarks to the Author):

The manuscript by Dudek and colleagues reports on unusual rectangular-shaped bacteria that divide along the transverse axis in the mouths of dolphins. The paper is interesting, very well-written, and logical. I agree with the general conclusions of the paper and the novel aspects of this organism(s). The electron microscopy is very well done and interpreted and a strength of the paper.

An unsatisfying aspect of the paper is obviously that the organism isn't identified. The authors did make concerted effort to do this, including (i) identification of common 16S rRNA genes from older data; (ii) FISH using prokaryote vs eukaryote probes; (iii) micromanipulation, followed by MDA, metagenomic binning, and taxonomic identification. The results of these analyses are logically described and conservatively

interpreted. However, I think there is a relatively simple path that may solve this. Figure 4 suggests these may be novel members of the Gammaproteobacteria and this could be addressed using class-specific FISH probes. If those would be successful, then FISH could be done using novel probes specific to the possible taxon. If the organism could be identified, then probably the genome could be interrogated to greatly enrich the value of the work.

Some other minor comments:

- The abstract only mentions genomics in the last sentence. Maybe this is appropriate given that the organism is unidentified but mentioning it in the last sentence only seems imprudent.

- Line 107: Several isolates of *Simonsiella* exist.

- The paper deals mostly with the long DAPI band morphotype. Can this be clarified a little more throughout the paper just to make this absolutely clear (for example, the FISH, 16S, cryo-EM, and micromanipulation sections)? Part of the reason I say this is that the short DAPI band morphotype is very similar to *Simonsiella*, so I wonder whether the short DAPI band morphotype is *Simonsiella* (or a relative) and the long DAPI band morphotype is something different. Maybe this can be addressed directly?

- Line 166: Note that *Simonsiella* is poorly studied and existing work suggest it may not be monophyletic (<https://pubmed.ncbi.nlm.nih.gov/12148653/>). I think the "net" should be opened to members of the Neisseriaceae, although the overall body of work still argues against the long DAPI band morphotype being a relative of *Simonsiella*.

- Line 231-236 seems to drop off without any clear conclusion on the possible identity of the long DAPI band organism. I suggest a clear sentence or two concluding this section. It seems like novel Gammaproteobacteria might be the most likely candidate? But I do advocate for caution here, as the authors have been in the current draft.

- Line 285-287: I don't understand what this text is discussing. The features in the RCUs aren't known, so how would we know that they're essential for interactions with the environment? I think it's probably true, but I suggest more conservative language here. Also, the preceding sentence concludes about the probably S-layer, so there's an unclear transition to the pili-like structures here, I think.

- Line 289: A shared S-layer or similar structure is interesting. It might be worth mentioning here that *Dictyoglomus* has a shared outer membrane (<https://pubmed.ncbi.nlm.nih.gov/22824581/>). It's obviously a different structure but there's some similarity there in terms of a microcolony sharing a common surface structure.

- Line 297-301: Was there a particular location in the dolphin mouth where either RCU morphotype was found? Note that *Simonsiella* is typically on the palate (for example: <https://pubmed.ncbi.nlm.nih.gov/886011/>) and many oral microbes have a preferred habitat.

- As mentioned above, FISH provides a straightforward route to identify the cells. Since FISH is working well for your cells, this is an obvious route. *Haloquadratum* provides a good example (e.g., <https://pubmed.ncbi.nlm.nih.gov/11207773/>)

- Figures: I would prefer the scale bars in figures to show the size of the bar so I could see cell size at a glance. Same comment for phase-contrast, FISH, and EMs.

-Reviewed by Brian Hedlund

Reviewer #4 (Remarks to the Author):

The manuscript by Dudek and colleagues describes previously uncharacterized bacteria discovered in the mouth of bottlenose dolphins. The authors used phase contrast and fluorescence microscopy combined sequencing to suggest that the observed object is of bacterial origin and is not the part of the *Simonsiella* genus. Interestingly, the bacteria had rectangle-like shape which gave them the name RCU. Further investigations with cryo electron microscopy and tomography showed peculiar appendages which were commonly observed next to the other bacteria as well as periodic density covering the surface of the RCU.

While the features spark curiosity, the paper is surprisingly descriptive; the organism is unknown, the features of the lifestyle of RCUs are not described, the molecular identity of the appendages and of the regular pattern on the surface of the RCU is unknown. In the current form the paper is definitely more suitable for a more specialized journal.

I was asked to comment on the cryo-EM part of the manuscript. Overall, it is well done with a few minor comments:

- 1. Scale bars in figure 5D,E don't seem to be 1 um long, please correct**
- 2. Tomograms in Figure 6 have sharp contrasty edges originating from tomographic reconstruction. They can be removed by normalizing the original tilt series to zero mean, softening their edges. Alternatively, as they don't contain useful information, the final images could be just cropped.**
- 3. Figure 7: panel D has inverted contrast while all the other images – not leading to representation inconsistencies.**

Dudek NK et al. “Previously uncharacterized rectangular bacterial structures in the dolphin mouth”

Responses to Reviewers (responses in **bold** type)

Reviewer #1 (Remarks to the Author):

This study describes an uncultivated microbe with unusual rectangular morphology examined with different imaging approaches. This examination of RCUs revealed distinct organization of DNA and membranes within the rectangularly shaped structure, suggesting RCUs were likely multicellular structures whose members divide along a transverse axis. CryoEM and cryoET revealed oddly shaped structures similar to pili. Attempts were also made to identify the phylogenetic identity of the microbe and sequencing its genome, but clearly linking a phylogenetic assignment to the RCUs was not possible.

This is an entirely descriptive paper, which is ok when describing such a morphologically interesting system, and there was a large amount of work that went into imaging and attempting to isolate and/or sequence the RCUs. I do not have any serious concerns about the data and methods presented. Unfortunately, the description seems incomplete, and we only have a partial story.

We appreciate the positive comments of the reviewer and have performed extensive new experiments to provide more information about the phylogenetic assignment of the “RCUs”. We acknowledge that the description remains incomplete and have highlighted intrinsic challenges for the application of genomics to organisms such as these.

It's not clear if RCUs are unicellular polyploid cells or multicellular structures, and the manuscript language seems to alternate referring to individual RCUs as 'cells' rather than aggregates, etc.

We have carefully reviewed and edited the manuscript to ensure consistency around reference to the RCUs (now called “rectangular bacterial structures” (RBSs)) as aggregate structures and not as single cells.

The data point towards the latter, but this could be more thoroughly addressed with additional staining and imaging, e.g., with a membrane stain like FM 1-43 and confocal microscopy. Or perhaps this can be addressed with a better explanation and illustration

of the gram staining and cryoEM to more clearly show the RCUs are composed of clearly separate cells.

We performed the suggested FM staining experiment. We first acquired fresh dolphin oral swab samples from our collaborators at the USN Marine Mammal Program in San Diego, CA, in the hope that more recently collected RBSs might be more accessible to FM dyes compared to older samples that had been in long-term storage at -80°C. Using these new samples, we:

- 1. Applied FM 4-64 dye (ThermoFisher Scientific Cat# T13320) directly to the dolphin oral swab samples. The FM dye did not stain any part of the RBSs.**
- 2. Launched renewed attempts to culture RBSs from these newly acquired swabs using media that supports the growth of other Betaproteobacteria (see other notes below about phylogenetic assignment), which might then have better stained using FM dyes; these efforts were unfortunately unsuccessful (see manuscript methods for more details).**

In the second paragraph of Results section “CryoEM reveals nanoscale surface and internal structures of RBSs”, we have added a more detailed discussion of the evidence that RBSs are composed of separate cells. This evidence consists of the following observations:

- 1. Around each segment there appears to be a plasma membrane, periplasmic space, and outer membrane, consistent with the proposal that each segment is an individual bacterial cell, with a cell wall structure suggestive of a Gram-negative bacterium. This observation is consistent with our Gram-staining of RBSs. In contrast, there is no such set of membranes surrounding each RBS as a whole (see Figures 5D, 6A,B (right), 7A).**
- 2. Observations of what appears to be cell division strongly support the proposal that segments are replicating individual cells, rather than membrane-bound chromosomes:**
 - a. In Figure 1G, neighboring DNA band pairs in an RBS form “H”-like shapes (white arrow), which we think reflect nucleoids (DNA bands) segregating in a cell undergoing longitudinal division.**
 - b. The RBSs appear to divide/split, such that 2-4 segments break off from the others. For example, in Figure 5A we can see sets of 2-3, 4, and 4-5 segments (as seen from left to right) that appear to be only marginally attached to one another. If the RBS were the cellular unit, these observations would imply a form of asymmetric cellular division with different quantities of genetic material forming**

daughter cells; while not impossible, such behavior would be highly unusual.

3. Even though a recent report describes the first discovery of a bacterium in which DNA is stored within a membrane (Katayama et al, 2019), the number of known bacterial species in which DNA is physically segregated from the cytoplasm, let alone organelle-bound, is extremely low. In contrast, the formation of multicellular bacterial structures has been documented in other bacterial species (reviewed in Lyons & Kolter, 2015).

In addition to adding additional discussion of these observations to the manuscript, we have also added annotations to Figure 5D that highlight features of the inner and outer membrane that surrounds each segment/cell.

Katayama, T., et al. "Membrane-bounded nucleoid discovered in a cultivated bacterium of the candidate phylum 'Atribacteria'." *bioRxiv* (2019): 728279.

Lyons, N.A., Kolter, R. "On the evolution of bacterial multicellularity." *Current Opinion in Microbiology* 24 (2015): 21-28.

In addition, RCUs ultimately were not assigned to a phylogenetic group or genome. I realize the difficulty, but more could be done. For example, the authors identify a limited number of likely genomes that belong to Bacteroidetes, Gammaproteobacteria, and Epsilonproteobacteria groups. FISH worked well with Eub338 probes and perhaps it will also work with probes for these groups, or possibly with probes at a finer phylogenetic scale without the authors needing to design and validate new probes.

We thank the reviewer for this suggestion. We have performed additional FISH experiments. These experiments suggest that RBSs are members of the Class Betaproteobacteria. We note that while members of the Betaproteobacteria genera *Alysiella*, *Simonsiella*, and *Conchiformibius* exhibit somewhat similar morphology, we performed extensive experiments to rule out the possibility that RBSs are affiliated with these genera. Results are as follows and have been integrated into the manuscript:

- Using an *S. muelleri*-specific cultivation medium, we were unable to culture the RBSs, despite acquiring new oral swab samples, while *S. muelleri* strain ATCC29453 (positive control) grew under the same conditions.
- 16S rRNA gene amplicon sequencing of the oral swabs containing RBSs did not yield any sequences corresponding to *S. muelleri* or other members of the Neisseriaceae (Betaproteobacteria Family that includes *S. muelleri*).

- Spike-in experiments with *S. muelleri* cells and our oral swab samples successfully yielded 16S rRNA gene amplicons with sequences identical to those of *S. muelleri*.

Finally, there was no connection between genomes and structures. For example, did any MAGs or unassembled reads have any similarity to genes involved in pili formation or similar structural elements?

New work published by Nyongesa *et al.* in August 2022 (below) suggests that the acquisition of *amiC2* was likely an important factor in the evolution of multicellularity and longitudinal division in the Neisseriaceae genera *Alysiella*, *Simonsiella* and *Conchiformibius*. Therefore, we examined our genome bins for putative *amiC2* genes, and detected *amiC2* in bins 4 (Epsilonproteobacteria, Arcobacter), 7 (Actinobacteria, Corynebacteriaceae), 9 (Bacteroidetes, Porphyromonas), 10 (Gammaproteobacteria, Gammaproteobacteria), 11 (Bacteroidetes, Tenacibaculum), 12 (Bacteroidetes, Flavobacteriaceae), and 16 (Betaproteobacteria, Alcaligenaceae). These results have been added to the manuscript.

We have the following concerns about pursuits into a line of inquiry as to whether genes involved in expression of pili or other novel structural features were present in bins, for the following reasons:

1. Given the genetic diversity that underlies known bacterial structures, we are not confident in our ability to recognize the genetic basis of novel structural elements, such as those involved in individual cells that form a rectangular structure.
2. Because of the novel appearance of the pili-like appendages on RBSs, we should not assume that they are indeed pili – and hence encoded by *pil* or associated genes. For example, Moissl *et al.* (2005) discovered and characterized novel pili-like appendages in archaea, named “hami” (plural) / “hamus” (singular). A follow up study by Perras *et al.* (2015) found that the “hamus subunit proteins, which are likely to self-assemble due to their predicted beta sheet topology, revealed no similarity to known microbial flagella, archaella, fimbria, or pili proteins, but high similarity to known S-layer proteins of the archaeal domain at their N-terminal region”.
3. We are not certain which (if any) genome bin corresponds to RBSs.
4. Even if we do assume that the pili-like appendages are encoded by typical *pil* genes and that our metagenomic sequencing data include RBS genome sequences, we run into two issues: Our bins correspond to partial genomes (some are quite incomplete), making it impossible to rule out the

presence of typical *pil* genes if an incomplete genome lacks them. Conversely, many genomes in nature encode pili, and so the finding of them on a genome contig may not be significant. Finally, the presence of known bacterial appendage genes in a genome bin does not necessarily suggest that the latter do or do not correspond to an RBS since bacteria can express multiple types of appendages.

Nonetheless, we performed the suggested analysis and investigated whether bins include genes involved in pilus and fimbriae systems. We submitted the protein sequences encoded by each bacterial genome bin to the KAAS annotation server. We then searched annotated bins for proteins in the “Pilus system” KEGG BRITE pathway (see https://www.kegg.jp/kegg-bin/show_brite?ko02035 > Pilus system) and recorded those that are involved in twitching motility, chemosensory pili systems, positive phototactic motility, pilus assembly, or fimbrial assembly. KEGG orthology identifiers for relevant proteins detected per genome bin are provided below. These results have not been included in the manuscript due to the concerns presented above.

Bin	Phylum	Lowest taxonomic ID	Twitching motility proteins (n = 10)	Chemosensory pili system proteins (n = 5)	Positive phototactic motility proteins (n = 5)	Pilus assembly proteins (n = 25)	Fimbrial proteins (n = 12)	Bin completeness (% CheckM)
1	Metazoa	s_Human	N/A	N/A	N/A	N/A	N/A	0.93
2	Gracilibacteria	p_Gracilibacteria	K02669	-	-	K02652, K02653, K02654	-	67.55
3	Epsilonproteobacteria	f_Campylobacteraceae*	K02669	-	-	-	-	13.79
4	Epsilonproteobacteria	g_Arcobacter	-	-	-	K02652, K02654	-	33.62
5	Gracilibacteria	p_Gracilibacteria	-	-	-	K02652	-	18.42
6	Fusobacteria	g_Fusobacterium	-	-	-	K02653	-	58.56
7	Actinobacteria	f_Corynebacteriaceae*	-	-	-	-	-	1.75
8	Fungi	c_Malasseziomycetes	N/A	N/A	N/A	N/A	N/A	0.16
9	Bacteroidetes	g_Porphyrromonas*	-	-	-	-	-	0
	Gammaproteobacteria	p_Gammaproteobacteria	K02657, K02660, K02667, K02668, K02669, K02670, K06596	-	-	-	-	
10				K06596		K02650, K02676		29.31
11	Bacteroidetes	g_Tenacibaculum*	-	-	-	-	-	76.03
12	Bacteroidetes	f_Flavobacteriaceae	-	-	-	-	-	89.67
	Gammaproteobacteria	f_Moraxellaceae*	K02657, K02659, K02660, K02667, K02668, K02669, K02670, K06596	-	-	K02650, K02652, K02653, K02654, K02656, K02676	-	57.94
13				K06596				
14	Fusobacteria	g_Oceanivirga	-	-	-	-	-	6.9
15	Gammaproteobacteria	p_Gammaproteobacteria	-	-	-	-	-	0
16	Betaproteobacteria	f_Alcaligenaceae	-	-	-	-	-	41.77
17	Gammaproteobacteria	g_Pasteurella*	-	-	-	K02662, K02663, K02664, K02666	-	12.07
	Gammaproteobacteria	p_Gammaproteobacteria	K02657, K02658, K02667, K02668, K02669, K02670, K06596	-	-	K02650, K02652, K02653, K02654, K02656, K02676	-	82.76
18				K06596				

Nyongesa, S., et al. "Evolution of longitudinal division in multicellular bacteria of the Neisseriaceae family." *Nature Communications* 13.1 (2022): 1-18.

Moissl, C., et al. "The unique structure of archaeal 'hami', highly complex cell appendages with nano-grappling hooks." *Molecular Microbiology* 56.2 (2005): 361-370.

Perras, A.K., et al. "S-layers at second glance? Altiarchaeal grappling hooks (hami) resemble archaeal S-layer proteins in structure and sequence." *Frontiers in Microbiology* 6 (2015): 543.

I think more definitely establishing any of these points would probably be enough to complete the story and make it a publishable unit, but I'm afraid that that manuscript does not meet that threshold yet in my opinion

As suggested, we have...

- included an extensive discussion of imaging data suggesting that RBSs are multicellular rather than polyploid structures;
- performed additional FISH experiments providing additional insight into the identity of the RBSs;
- included a search within our sequencing data for *amiC2*, a gene which has been shown to play a role in the evolution of multicellularity and/or longitudinal division in members of the *Neisseriaceae* that possess these properties.

We have additionally...

- expanded the 16S rRNA gene amplicon survey of dolphin oral samples. Originally, we sequenced three samples that were visually confirmed to contain RBSs via microscopy; we now have deep sequencing data from 54 dolphin oral samples. Of 48 screened for RBSs using a high-throughput phase contrast microscopy protocol, 22 (45.8%) were confirmed to contain RBSs via microscopy. Rarefaction curves show that samples were sequenced to near-saturation, and yet no *Simonsiella* or *Neisseriaceae* were detected in any samples (see revised Figure 2). Spike-in experiments with *S. muelleri* cells and aliquots of dolphin oral samples resulted in the identification of *S. muelleri*, as expected (positive control);
- used high-throughput phase contrast microscopy with 25 samples from specific locations in the dolphin oral cavity (palate, gingiva, and buccal surface) to assess RBS habitat preference. We additionally provide better

characterization of the prevalence of each RBS morphotype across the full set of 73 samples for which imaging data were obtained.

- performed additional cryoEM/ET experiments to characterize the RBSs. More specifically, we used cryogenic focused ion beam (FIB) scanning electron microscopy (SEM) (Marko *et al.* 2007; see reference below) to mill thinner lamellae from RBS samples, as we have successfully done in the past with yeast cells (Wu *et al.* 2020; see reference below). However, the physical properties of the RBSs were not amenable to this type of analysis, despite our best efforts. We report these efforts in the Results section and provide estimates for RBS-As thickness, corresponding details concerning cryoFIB-SEM in the Methods section, and considerations regarding the challenges of imaging specimens of this thickness in the Discussion. Unfortunately, the thickness seemed to be too much for cryoET, but on the other hand, not thick enough for cryoFIB-SEM.
- performed new culture-based experiments in which *S. muelleri* was successfully cultured yet RBSs failed to grow under the same conditions. This suggested that the physiological requirements of RBSs and *S. muelleri* are fundamentally different and argues against them being the same species.
- added discussion comparing the visual appearance of RBSs against those of *Alysiella*, *Simonsiella* and *Conchiformibius*, courtesy of new fluorescence and electron microscopy imaging data of these taxa presented in Nyongesa *et al.* (2022).
- performed new phylogenetic analyses to confirm that our Betaproteobacteria bin corresponds to an organism affiliated with the family *Alcaligenaceae*, given that FISH suggested a Betaproteobacteria identity for the RBSs; it likely represents a previously uncharacterized genus.

Nyongesa, S., *et al.* "Evolution of longitudinal division in multicellular bacteria of the Neisseriaceae family." *Nature Communications* 13.1 (2022): 1-18.

Marko, M., Hsieh, C., Schalek, R., Frank, J. and Mannella, C. Focused-ion-beam thinning of frozen-hydrated biological specimens for cryo-electron microscopy. *Nature methods*, 4(3) (2007): 215-217.

Wu, G.H., Mitchell, P.G., Galaz-Montoya, J.G., Hecksel, C.W., Sontag, E.M., Gangadharan, V., Marshman, J., Mankus, D., Bisher, M.E., Lytton-Jean, A.K. and Frydman, J. Multi-scale 3D cryo-correlative microscopy for vitrified cells. *Structure*, 28(11) (2020): 1231-1237.

We appreciate the comments and suggestions from this reviewer and believe that our analyses and manuscript have benefitted from them. We are happy to respond to any further questions and comments that you may have.

Reviewer #2 (Remarks to the Author):

Because bacterial cell biology has been focusing on model bacteria there is a tremendous need for investigations such as this one conducted by Dudek et al. In spite not being able to isolate them in pure culture, the authors discovered that:

- 1) multicellular, likely longitudinally dividing bacteria (of unknown phylogenetic placement) may inhabit the dolphin mouth
- 2) each multicellular unit of longitudinally dividing bacteria contains multiple stripes of DNA that can be shorter or longer (not quantified)
- 3) they have pili-like structures that are splayed out.

The most remarkable result of this study, in my opinion, is the discovery of pilus-like appendages which protrude as bundles of threads partially parallel to each other but splayed-out at the tips (number 3). This is, to the best of my knowledge, unprecedented.

We appreciate these comments of the reviewer.

There are however a number of claims/conclusions which are not supported:

- 1) Already in the title and throughout the manuscript the authors claim to have characterized "rectangular" bacteria: although the multicellular units might appear rectangular, what are the evidence(s) that each individual bacterium is rectangular, i.e. each cell has 90 degrees, non-rounded corners? When I look at the beautiful cryoEM-based 3D reconstructions, the individual cells look rod-shaped to me. If, with "rectangular", the authors refer to the filament, then, perhaps, Rectangular Multicellular Unit (RMU) would be more appropriate than Rectangular Cell-like Unit (RCU).

We appreciate this feedback and agree that the descriptive title for the structures should be improved.

The original term, “RCU” referred to the overall structure, which appeared to consist of multiple cells, having a rectangular appearance. We agree that the individual cells appear rod-shaped.

We propose a revised term, “Rectangular Bacterial Structures” (RBSs) in lieu of RCUs or RMUs. While our evidence strongly suggests that the filaments/segments are individual cells and that the structures in question are multicellular (see response to Reviewer #1), we prefer to err on the side of caution and avoid using a name that assumes this to be fact, until it has been proven.

The wording throughout the paper has been modified such that “RCU” is now replaced with “RBS”.

2) The authors claim that the "rectangular" units are not belonging to the genus *Simonsiella* because this genus was not detected in all their sequencing efforts: was the sequencing depth sufficient to detect *Simonsiella* even if it had been present in low abundance? The authors need to show rarefaction curves to support their claim/to make sure that the sequencing was deep enough to discover rare taxa.

Thank you for highlighting this important issue. We have addressed it in the following fashion:

- 1. To increase the robustness of our analysis, we performed deep 16S rRNA gene amplicon sequencing of 54 additional dolphin oral samples. Of these, 48 were screened for RBSs via microscopy and 22 (45.8%) were confirmed to contain them. We achieved a median read depth of 92,077 reads per sample (min: 50,772; max: 265,108).**
- 2. We then plotted rarefaction curves for all samples and showed that for the majority of our samples, we reached near-complete ASV detection. We found no *S. muelleri* (or Neisseriaceae) amplicons. We have revised Figure 2 to include rarefaction curves for each sample (panel A) and the relative abundances of Betaproteobacteria Families in samples with members of these Families (panel B).**

Alternatively, failure in amplifying *Simonsiella* 16S rDNA by qPCR or PCR with *Simonsiella*-specific primers is needed to support the claim that "the rectangular units are distinct from *Simonsiella*".

Another possibility is to perform FISH with *Simonsiella* (or Neisseriaceae) 16S rRNA gene-specific probe(s).

To the best of our knowledge, there are no *Simonsiella*-specific primers available (e.g., in ProbeBase). Designing and validating a *de novo* probe that is sufficiently specific and sensitive for a single taxonomic group seemed inefficient and unnecessary.

Instead, we prepared aliquots of dolphin oral samples that were visually confirmed to contain RBSs (but for which no *S. muelleri* ASVs were detected) and spiked-in *S. muelleri* strain ATCC29453 cells. Based on 16S rRNA gene amplicon sequencing, we successfully identified *Simonsiella* reads from the spike-in sample.

Thus, the lack of *S. muelleri* reads in any of the oral swab samples (n = 54) sequenced suggests that the RBSs are unlikely to be *S. muelleri*.

We have provided additional new experimental data supporting the conclusion that RBSs are not Neisseriaceae, including *S. muelleri*. The following text has been added to the Discussion section:

Results collected here strongly suggest that the RBS-As are not affiliated with the Neisseriaceae family, which includes three known genera (*Simonsiella*, *Alysiella*, and *Conchiformibium*) whose members also form rectangularly-shaped clusters of rod-like cells. First, the marker gene amplicon sequencing workflow employed here did not detect any Neisseriaceae amplicons in 54 dolphin oral samples that underwent deep 16S rRNA gene amplicon sequencing, although this workflow did detect *S. muelleri* after cells from this taxon were intentionally spiked into aliquots of dolphin oral samples. Second, attempts to culture RBA-As using techniques that were employed successfully to culture *S. muelleri*, failed, suggesting that the RBS-As have different physiological requirements for growth than those required by *S. muelleri*. Third, no Neisseriaceae genomes were recovered from the mini-metagenomics experiment. Fourth, visual comparisons of RBS-A images presented here and the TEM and fluorescence microscopy images of Neisseriaceae members (*Alysiella*, *Simonsiella*, and *Conchiformibius*) presented in Nyongesa et al. (Figs. 2, 4, Supplementary Fig. 3) suggest that RBSs are not affiliated with these taxa. For example, RBSs appear to contain segments (cells) that are embedded in a matrix-like material and fully encapsulated by an S-layer-like structure, whereas the longitudinally dividing Neisseriaceae imaged by Nyongesa et al. have cells that appear to be less “joined” as a unit.

Additional comments:

line 29: what do the authors mean by "microbial structure"? Please clarify

We have removed the term “microbial structure” and reorganized the sentence to improve clarity.

line 30, 134: what is rectangular: the cell or the filament? If the authors are referring to the individuals cells, these/their corners should be measured

We apologize for the confusion: the term “rectangular” refers to the unit as a whole - not the cells/segments/filaments. We have addressed this point as part of major concern #1 above and fixed the wording in line 134.

line 31: please specify that the genomic DNA underwent an amplification step.

We have specified on what was formerly line 31 that genomic DNA underwent an amplification step.

line 33: may be distinct from *Simonsiella* (see my major concerns above).

We have added additional evidence in this version of the manuscript that supports the conclusion that the RBSs are not *S. muelleri*. This line now reads as follows (and uses the word “suggest” – we do not make definitive statements about the identity of the RBSs):

Multiple lines of evidence suggested that RBSs are bacterial and distinct from the Neisseriaceae genera *Alysiella*, *Simonsiella*, and *Conchiformibius*, with which they share in some cases similar morphology and division patterning, including genomic DNA sequencing of micromanipulated RBSs, 16S rRNA gene amplicon sequencing, and fluorescence *in situ* hybridization.

More detail on those lines of evidence (discussed in the Discussion section) are provided above.

Nyongesa, S., et al. "Evolution of longitudinal division in multicellular bacteria of the Neisseriaceae family." *Nature Commun* 13.1 (2022): 1-18.

line 36: "membrane-bound segments": please clarify.

We have clarified the wording to make explicit our belief that each membrane-bound segment is an individual cell. Without definitive evidence to support this point, however, we are intentionally cautious about the wording. The sentence now reads:

“Cryogenic transmission electron microscopy and tomography revealed parallel membrane-bound segments, suspected to be cells, [...]”

line 45: the authors should provide evidence for the existence of an S layer.

We provide evidence in the manuscript that RBSs are encapsulated by a periodic surface layer reminiscent of an S-layer. In the manuscript text, we have been careful to avoid claiming that the periodic surface layer is an S-layer and rather refer to it as "S-layer-like".

In our keywords, however, we include the term “S-layer” since it is a more useful search term than one of many vague descriptive alternatives such as “uncharacterized periodic surface layer” or “S-layer-like structure”. For a reader interested in S-layer-like structures, “S-layer” is likely to be the most common and useful search term for such subject material.

line 103: not only *Candidatus T. hypermnestrae*, but also the fruit-fly endosymbiont *Spiroplasma poulsonii* (Ramond et al., 2016) and *Candidatus T. oneisti* divide longitudinally (Leisch et al., 2012)

Thank you for providing these additional examples and references. We have incorporated them into the manuscript.

line 106: up to now, only a species of *Simonsiella* is known, *S. muelleri*.

We have clarified this point in the text and added discussion of members of the genera *Alysiella* and *Conchiformibium* to the introduction.

line 108: *Simonsiella muelleri* is cultivable (see for example Hedlund BP, Tønnum T. *Simonsiella*. Bergey’s Man Syst Archaea Bact [Internet]. 2015;1–12).

This was a typo - thank you for pointing it out. It has been corrected.

line 134: multicellular units, not individual cells, contain multiple bands

Agreed - we have revised the phrasing.

line 139: we understand that they are prevalent in dolphin mouths (they were found in 20 out of 28) but can the author provide an estimate of their relative abundance?

To address this question, we conducted high-throughput imaging experiments to better characterize the prevalence of RBSs of each morphotype in 73 dolphin oral samples from eight dolphins. In these experiments, we used the Strain Library Imaging Protocol (SLIP) proposed by Shi et al., 2017 (below) to collect 226 fields of view (FOVs) per dolphin oral sample. Our approach was to count the number of RBS-As/Bs per FOV, estimate the average number of bacterial cells per set of FOVs, and then use this information to estimate the relative abundance of RBS-As/Bs. We ultimately abandoned this attempt, however, due to technical properties of the dolphin oral samples, which are derived from complex environmental communities.

Unlike in an axenic culture, in which bacterial cells are of a similar size and can be observed equally well, there are diverse bacterial morphologies and sizes in the dolphin oral samples (including some bacteria that are very small) and an uneven distribution of bacterial cells. As an example, below are two FOVs obtained from the same sample. The right panel is annotated with green circles around one or more putative bacterial cells (although there may be more); the left is unannotated to provide a point of comparison. Many putative bacterial cells are small, making it inherently challenging to count them accurately. It is also challenging to distinguish between bacterial cells and microscopy artifacts and/or debris.

As an example, here is a field of view; unannotated and the annotated with green circles indicating potential bacterial cells:

For comparison, a different field of view (note that the comparison across FOVs is essential to detect microscope artifacts):

Even if one were to invest the many needed hours into assessing the number of presumed bacterial cells in the samples, one is still left with the problem that the density of bacteria per FOV is not uniform, due to the nature of the communities themselves (samples were thoroughly vortexed before being applied to slides for microscopy). The image below illustrates the nature of the problem:

Furthermore, in some cases it is challenging to determine what constitutes a single cell. For example:

Due to these challenges inherent to the nature of working with complex environmental communities, we believe that estimates of the relative abundance of RBSs are unlikely to be reliable or reproducible (in fact, two members of our group independently made attempts to establish reliable counts using the same imaging data and the results were dismayingly discordant).

Instead, we have added a Supplementary Table that provides an order of magnitude estimate of counts of each identified morphotype. We found that RBSs of morphotype A, henceforth referred to as RBS-As, were detected in 39/73 samples from 7/8 dolphins (note that the number of samples per dolphin is not constant and that different oral locations were swabbed in different years). RBSs of morphotype B, henceforth referred to as RBS-Bs, were detected in 42/73 samples from 6/8 dolphins. Of 25 of these 73 samples that were collected from distinct oral sites, RBSs were detected in palatal ($n_{\text{RBS-A}} = 5/5$; $n_{\text{RBS-B}} = 4/5$) and gingival samples ($n_{\text{RBS-A}} = 12/15$; $n_{\text{RBS-B}} = 14/15$), but less frequently in buccal samples ($n_{\text{RBS-A}} = 0/5$; $n_{\text{RBS-B}} = 2/5$). Thus, we infer that the RBSs are stable colonizers in the dolphin oral cavity and have preferred colonization locations.

Shi, H., Colavin, A., Lee, T. K. & Huang, K. C. Strain Library Imaging Protocol for high-throughput, automated single-cell microscopy of large bacterial collections arrayed on multiwell plates. *Nat Protoc* 12, 429-438, doi:10.1038/nprot.2016.181 (2017).

line 141: do the authors mean a stack of squared pancakes viewed from the side?

This statement has been reworded to improve clarity. It now reads:

“The RBSs resembled rectangular prisms; they are not cylindrical.”

line 148: "the individual DAPI-stained bands tended to be longer": how much longer? Can the authors please specify?

We have added summary statistics to the manuscript detailing the difference in length and width between the two RBS morphotypes and added a reference to Fig. 1h, which shows the distribution of RBS dimensions for each morphotype.

The dimensions of the RBSs are as follows:

Morphotype A:

Median length: 3.95 ± 2.89 μm median absolute deviation (MAD)

Median width: 5.08 ± 0.10 μm MAD

$n = 15$ RBSs

Morphotype B:

Median length: 3.08 ± 0.93 μm MAD

Median width: 2.21 ± 0.56 μm MAD

$n = 8$ RBSs

Using Student's t-test, median length and width are both significantly different between the two morphotypes. For length, $p = 0.03$, and for width, $p = 10^{-8}$.

Note that the dimensions of RBSs were not normally distributed and thus we used median and median absolute deviation.

line 152: representative image (D,E) shows an RCU that is longer than the RCUs in (B,C). Or do the authors refer to the single cells?

We refer to the length of the presumed DNA bands. To improve clarity, we have replaced the names for the different morphologies from “long DAPI band” and “short DAPI band” with “morphotype A” and “morphotype B”.

line 153, line 156: can the authors clarify/estimate the occurrence of each morphotype?

As described above, we have surveyed 73 dolphin oral samples using a high-throughput, automated imaging workflow. We counted the number of RBSs of each morphotype in the resulting imaging data, offer a high-level summary in the Results section, and present the full results in Supplementary Table 1.

line 163: neither *Candidatus T. hypermenestrae* cells nor *Simonsiella* cells/filaments, based on the literature, are rectangular (Hedlund BP, Tønjum T. *Simonsiella*. Bergey's Man Syst Archaea Bact [Internet]. 2015;1–12). They all are considered rod-shaped.

We agree that *Candidatus T. hypermenestrae* and *S. muelleri* have rod-shaped cells. In the sentences that precede and span line 163 we state:

“[RBSs] bear qualitative resemblance to [*S. muelleri*], which consist of rod-shaped bacteria that collectively form a rectangular unit and are oral commensals in mammals. Thus, we first sought to evaluate the likelihood that [RBSs] are members of this genus.”

line 169: *Simonsiella muelleri* or *Simonsiella* sp.?

We have added the following clarification to the manuscript:

“A putative *S. muelleri* OTU was detected in the Sanger library from the mouth and gastric fluid of a sea lion examined in the same amplicon study (NCBI accession number JQ205404.1); this partial 16S rRNA gene sequence has 94.58% identity over 99% query cover to the partial *S.*

***muelleri* ATCC 29453 16S rRNA gene sequence (NCBI accession number NR_025144.1)”**

line 170: the fact that it worked on sea lion samples is no guarantee that it works on dolphin samples

We provided the sea lion example as evidence that the DNA extraction protocol was capable of lysing *S. muelleri* cells and the bioinformatic pipeline used in the former study was capable of identifying *S. muelleri*. For example, our past experience suggests that *S. muelleri* does not require bead beating to lyse its cell wall. Both this and the former study by our group used identical extraction and amplification procedures.

To provide greater clarity on this point, however, we performed a spike-in experiment where *S. muelleri* cells were spiked into aliquots of dolphin oral samples that previously underwent 16S rRNA gene amplicon sequencing and had no *S. muelleri* detected. *S. muelleri* amplicons were detected in these aliquots, indicating that the wet lab and bioinformatics pipelines employed for our 16S rRNA gene amplicon survey were capable of detecting *S. muelleri*.

line 197: the authors did not employ single-cell genomics. Single-cell genomics refers to the process of separating communities into single units, the amplification of DNA of single cells and its subsequent sequencing. Instead, the authors micromanipulated the communities via suction and, most likely, amplified DNA of multiple cells (the “negative” samples reflect this).

We agree and have changed the term “single-cell genomics” to “mini-metagenomics” to reflect that more than one cell (as well as potentially cell-free DNA) was sequenced.

The term “mini-metagenomics” was coined, to the best of our knowledge, by Yu et al 2017 (below). It refers to an approach in which a small number of cells are subsampled from a complex community and their DNA is amplified using multiple displacement amplification.

Yu, Feiqiao Brian, et al. "Microfluidic-based mini-metagenomics enables discovery of novel microbial lineages from complex environmental samples." *Elife* 6 (2017): e26580.

line 198: In principle yes, but, in fact, the phi29 polymerase - commonly used for amplification - has a strong bias towards sequence composition, i.e. GC-rich stretches are usually not well amplified, and hence, are underrepresented in sequencing.

We agree that GC-rich DNA sequences may experience amplification bias, which is particularly problematic when attempting to characterize the composition of a community, since GC-rich genomes may be under-represented in sequencing results. The goal of our study, however, was to characterize any genomes that were present at all. The GC content of a genome would have to be extremely biased to make it completely undetectable. The question then becomes what counts as “extreme” GC content?

In our study, DNA amplification was performed using the REPLI-g kit. Qiagen evaluated the bias of the kit with regards to GC content, the results of which are summarized in Figure 2 in a white paper by Meier et al., 2014:

Figure 2. The REPLI-g Single Cell Kit ensures minimal bias, even for samples with high GC content. GC% bias was compared in NGS runs using non-amplified genomic DNA of approximately 150,000 *Bacillus subtilis* cells (red) and two independent REPLI-g Single Cell WGA DNA sets starting from a few cells of *Bacillus subtilis* (purple and blue squares). The x-axis describes the GC content (%) of 100 bp bins of the *Bacillus subtilis* genome and the y-axis shows the number of reads obtained in NGS, related to the number of expected reads. The purple and blue curves indicate the sequencing result of the genome from

The phi29 polymerase does have amplification bias on very low (<30%) and very high (>65%) GC content stretches of DNA, but readily amplifies DNA in the 30-65 GC% content range. It amplifies DNA with >70% GC content, although less well.

The GC content of *S. muelleri* and exemplar diatoms with sequenced genomes are as follows:

(Note: To obtain results for diatoms, we searched for “diatom” in the NCBI genome browser and supplemented those findings with a Google scholar search).

S. muelleri: 41.5% (NCBI genome browser)

Average GC content of diatom genomes, sorted by increasing GC content

Organism	Genome GC content	Source
Skeletonema costatum	45.10%	Ogura et al., 2018
Fistulifera solaris	46%	Tanaka et al., 2015
Thalassiosira pseudonana	46.90%	NCBI genome browser
Phaeodactylum tricorutum	48.80%	NCBI genome browser
Fistulifera pelliculosa	49.10%	NCBI genome browser
Thalassiosira oceanica	53.50%	NCBI genome browser
Chlamydomonas reinhardtii	63.10%	Scala et al., 2002

These taxonomic groups appear to have genomic GC content within the range of what will be amplified with minimal bias by the phi 29 polymerase in the REPLI-g kit. While some protein coding genes may have aberrant GC content and may be amplified less well, as a whole we would expect enough amplification of genomic DNA to obtain a signal in our mini-metagenomics experiment.

Meier, A. F. E., et al. "Genomic analysis of individual cells by NGS and real-time PCR." QIAGEN Scientific Article (2014): 1-8.

Ogura, A., et al. "Comparative genome and transcriptome analysis of diatom, *Skeletonema costatum*, reveals evolution of genes for harmful algal bloom." BMC genomics 19.1 (2018): 1-12.

Tanaka, Tsuyoshi, et al. "Oil accumulation by the oleaginous diatom *Fistulifera solaris* as revealed by the genome and transcriptome." The Plant Cell 27.1 (2015): 162-176.

Scala, S., et al. "Genome properties of the diatom *Phaeodactylum tricornutum*." *Plant physiology* 129.3 (2002): 993-1002.

lines 211-212: to be sure that the rectangular units are neither diatoms nor *Simonsiella* the authors should show that diatoms and *Simonsiella* cells can be successfully lysed and their DNA amplified by phi29.

We have modified this statement to read as follows:

“Notably, no *S. muelleri*, Neisseriaceae, or diatom genomes were recovered, although positive controls for these taxa were not included in the experimental design.”

Given that the spike-in experiment we conducted strongly suggests that *S. muelleri* should have been detected in dolphin oral samples, were it present, we opted not to add an *S. muelleri* positive control to the mini-metagenomics experiment. In addition to our results suggesting that *S. muelleri* lyses using standard protocols, Hedlund et al. (2002) lysed cells and amplified DNA from *S. muelleri* and other members of the Neisseriaceae using a standard, off-the-shelf kit (the Instagene kit from Bio-Rad) without requiring additional procedures to lyse the cell wall (e.g., bead beating) or ensure DNA amplification (e.g., removal of PCR inhibitors – although these will usually inhibit amplification of an entire sample, not just a single species within a sample).

Diatoms were not included as positive controls based on the following logic:

- Throughout the literature, some studies include an additional step to ensure lysis of the frustule of diatoms, while others do not. For example, Guo et al 2016 created a mock community of diatoms (sample act7) and performed both 18S amplicon sequencing (act7.1) and WGA sequencing (act7.2) on these samples and samples of dinoflagellates (other marine plankton). For WGA, isolation and amplification of DNA was performed using the same commercial kit used in our study (Repli-g from Qiagen).

They report:

“Comparing the sample [...] act7.2 (WGA samples) with [...] act7.1 [18S rRNA gene amplicon sequenced aliquots of the same samples], species annotated in these samples were the same but their abundance differed greatly”

(Notably, Guo et al. performed WGA after fixing samples with Lugol’s solution, which they hypothesize may have exacerbated the extent to which abundances differed.)

- **Given that diatoms may or may not require additional lysis, and without having a specific type of diatom on which to focus WGA efforts, we concluded that adding a positive control would be of limited value. Instead, we have clarified the wording in this section to emphasize the limitations of this approach in detecting diatom DNA.**

Hedlund, B.P., Staley, J.T. "Phylogeny of the genus *Simonsiella* and other members of the Neisseriaceae." International journal of systematic and evolutionary microbiology 52.4 (2002): 1377-1382.

Guo, L., Sui, Z., Liu, Y. "Quantitative analysis of dinoflagellates and diatoms community via Miseq sequencing of actin gene and v9 region of 18S rDNA." Scientific Reports 6.1 (2016): 1-9.

line 281-282: "such as the high and variable curvature of the regions with a continuous periodic surface covering": please clarify.

The paragraph in question has been modified to contain a more complete description of why the variable curvature of the regions with a continuous periodic surface covering pose a challenge to cryoEM/ET analyses. Specifically, it now reads:

“[...] such as the variable curvature of the regions with a continuous periodic surface covering. Successful subtomogram averaging typically relies on averaging identical structures, for example, repeated copies of a macromolecular complex, such as ribosomes in the same functional state. One can compensate for structural variability in the form of conformational and compositional heterogeneity with large datasets composed of thousands of subtomograms in combination with advanced classification methods. However, in our datasets, both the pili-like appendages and the S-layer-like surface feature were structurally heterogeneous and sparsely distributed in the RBS-As (e.g., the S-layer-like surface feature is discontinuous along the membrane of the RBS-As, and in the stretches where it is, it exhibits differential curvature) and thus were observed only in a fraction of our tomograms.”

line 304: in my opinion, the wording "single-cell" genomics is not appropriate (see above)

This point is addressed above - we have changed the wording to “mini-metagenomics” in the sentence in question.

line 320: on the basis of what do the authors think that the rectangular units, which thrive in the mouth of dolphins, an aerobic habitat, could be strictly anaerobe?

We have removed the sentence in question. With the exception of species living in deep periodontal pockets, we agree that it is unlikely that oral bacteria will be strictly anaerobic. The original wording was overly cautious.

We have also attempted to culture the RBSs under anaerobic conditions using fresh oral swab samples. While we were able to cultivate some bacteria, none of the colonies contained members with a morphology consistent with the RBSs.

line 327: *S. muelleri*, *Spiroplasma poulsonii*, *Ca. Toneisti* and *Ca. T. hypermnestrae*

We have added these additional example species to the discussion.

line 544: did the authors do negative controls?

Yes, 32 negative (PBS) controls were included in the DNA extraction, amplification, and sequencing steps of the 16S rRNA gene amplicon survey. Despite the absence of visible amplification bands, we sequenced the output of these PCRs, and obtained 628--9942 reads per control (median 1982 reads), in comparison to 50,772--265,108 reads per experimental sample (median 92,077 reads).

line 547: concentration of reagents and cycling conditions are missing

PCR was performed according to the manufacturer's instructions. This information has been added to the manuscript.

line 554: how many reads per samples?

We achieved a median read depth of 92,077 reads (min: 50,772, max: 265,108). This information has been added to the Methods section.

line 620: how do the authors search the metagenome assembly?

We have added the following methods description to the manuscript:

Profile HMMs for relevant proteins were obtained from the Pfam database (Finn et al., 2016) and queried against our dataset using HMMER Suite v. 3.1b2 (Eddy, 2011).

line 924: on the basis of what do the authors claim that the black dots correspond to storage granules or lipid droplets?

We have removed this comparison from the manuscript.

We appreciate the comments and suggestions from this reviewer and believe that our analyses and manuscript have benefitted from them. We are happy to respond to any further questions and comments that you may have.

Reviewer #3 (Remarks to the Author):

The manuscript by Dudek and colleagues reports on unusual rectangular-shaped bacteria that divide along the transverse axis in the mouths of dolphins. The paper is interesting, very well-written, and logical. I agree with the general conclusions of the paper and the novel aspects of this organism(s). The electron microscopy is very well done and interpreted and a strength of the paper.

An unsatisfying aspect of the paper is obviously that the organism isn't identified. The authors did make concerted effort to do this, including (i) identification of common 16S rRNA genes from older data; (ii) FISH using prokaryote vs eukaryote probes; (iii) micromanipulation, followed by MDA, metagenomic binning, and taxonomic identification. The results of these analyses are logically described and conservatively interpreted. However, I think there is a relatively simple path that may solve this. Figure 4 suggests these may be novel members of the Gammaproteobacteria and this could be addressed using class-specific FISH probes. If those would be successful, then FISH could be done using novel probes specific to the possible taxon. If the organism could be identified, then probably the genome could be interrogated to greatly enrich the value of the work.

We thank the reviewer for this suggestion. We have performed additional FISH experiments that suggest RCUs (now referred to as “RBSs” – see response to reviewer #2) are members of the Betaproteobacteria. We have incorporated these findings into the manuscript.

We note that while *S. muelleri* (another rectangular bacterium) is also a Betaproteobacteria, we performed extensive experiments to rule out the possibility that RBSs are *S. muelleri*. These results are synthesized in the Discussion section as follows:

Results collected here strongly suggest that the RBS-As are not affiliated with the Neisseriaceae family, which includes three known genera (*Simonsiella*, *Alysiella*, and *Conchiformibium*) whose members also form rectangularly-shaped clusters of rod-like cells. First, the marker gene amplicon sequencing workflow employed here did not detect any Neisseriaceae amplicons in 54 dolphin oral samples that underwent deep 16S rRNA gene amplicon sequencing, although this workflow did detect *S. muelleri* after cells from this taxon were intentionally spiked in to aliquots of dolphin oral samples. Second, attempts to culture RBA-As using techniques that were employed successfully to culture *S. muelleri* failed, suggesting that the RBS-As have different physiological requirements for

growth than those required by *S. muelleri*. Third, no Neisseriaceae genomes were recovered from the mini-metagenomics experiment. Fourth, visual comparisons of RBS-A imaging data presented here with the TEM and fluorescence microscopy images presented of Neisseriaceae members (*Alysiella*, *Simonsiella*, and *Conchiformibius*) presented in Nyongesa et al. (Figs. 2, 4, Supplementary Fig. 3) suggest that they are different taxa. For example, RBSs appear to contain segments (cells) that are embedded in a matrix-like material and fully encapsulated by an S-layer-like structure, whereas the longitudinally dividing Neisseriaceae imaged by Nyongesa et al., have cells that appear to be less “joined” as a unit.

Some other minor comments:

- The abstract only mentions genomics in the last sentence. Maybe this is appropriate given that the organism is unidentified but mentioning it in the last sentence only seems imprudent.

Our abstract now references the genomics work earlier. We have clarified the wording in this sentence to make it more apparent.

- Line 107: Several isolates of *Simonsiella* exist.

This was a typo - thank you for pointing it out. It has been corrected.

- The paper deals mostly with the long DAPI band morphotype. Can this be clarified a little more throughout the paper just to make this absolutely clear (for example, the FISH, 16S, cryo-EM, and micromanipulation sections)? Part of the reason I say this is that the short DAPI band morphotype is very similar to *Simonsiella*, so I wonder whether the short DAPI band morphotype is *Simonsiella* (or a relative) and the long DAPI band morphotype is something different. Maybe this can be addressed directly?

We appreciate this feedback and agree that this is an important point to emphasize. After simplifying the morphotype names to morphotype “A” and “B” (following feedback from another reviewer), we created a new abbreviation to distinguish the one studied in greatest detail here: RBS-A. We have updated the manuscript to refer primarily to RBS-As, rather than RBSs, in the Results, Discussion, and Methods sections. As we discuss below, we have provided additional evidence in our revision that *S. muelleri* is not present in our samples.

- Line 166: Note that *Simonsiella* is poorly studied and existing work suggests it may not be monophyletic (<https://pubmed.ncbi.nlm.nih.gov/12148653/>). I think the “net” should

be opened to members of the Neisseriaceae, although the overall body of work still argues against the long DAPI band morphotype being a relative of Simonsiella.

We appreciate this feedback and agree with the logic. We have expanded the net to include all members of the Neisseriaceae. In brief, both the former and present studies did not detect any members of the Neisseriaceae in dolphin oral samples, although they were detected in other types of samples, indicating that the DNA extraction protocol used in both the former and present studies was capable of extracting DNA from members of the Neisseriaceae. The manuscript wording has been modified accordingly.

- Line 231-236 seems to drop off without any clear conclusion on the possible identity of the long DAPI band organism. I suggest a clear sentence or two concluding this section. It seems like novel Gammaproteobacteria might be the most likely candidate? But I do advocate for caution here, as the authors have been in the current draft.

We have added the following, cautious statement to the end of the section in question:

“While no conclusive insights can be drawn about RBS-A identity, FISH results suggest a putative affiliation with the Betaproteobacteria class. We note that a Betaproteobacteria genome from the family Alcaligenaceae was recovered from the mini-metagenomics experiment; the affiliation of this bin with Alcaligenaceae was confirmed via phylogenetic analysis of the 16S rRNA and ribosomal protein S3 genes (Supplementary Fig. 4, 5, Supplementary Methods). An ASV consistent with the 16S rRNA gene of that bin is present in 11/13 of samples in which RBS-As were visually confirmed to be present and for which amplicon sequencing data are available.”

- Line 285-287: I don't understand what this text is discussing. The features in the RCUs aren't known, so how would we know that they're essential for interactions with the environment? I think it's probably true, but I suggest more conservative language here. Also, the preceding sentence concludes about the probable S-layer, so there's an unclear transition to the pili-like structures here, I think.

We have modified the language to be more clear and conservative. More specifically, we modified the second half of the paragraph to say, “Characterized cell surface features such as pili and S-layers play key roles in a cell's interaction with its environment^{35,37}. The nature of the pilus-like appendages and the S-layer-like periodic surface covering merit future investigation.”

With regards to the transition between the pili-like appendages and the periodic surface layer, the discussion of the former has now been substantially reduced. The opening paragraph begins with a discussion of both features, and the paragraph now discusses both equally.

- Line 289: A shared S-layer or similar structure is interesting. It might be worth mentioning here that Dictyoglomus has a shared outer membrane (<https://pubmed.ncbi.nlm.nih.gov/22824581/>). It's obviously a different structure but there's some similarity there in terms of a microcolony sharing a common surface structure.

Thank you for sharing this very interesting reference. We incorporated this reference into our discussion (as described above, we have modified line 289 such that discussion of the periodic surface layer is now limited to technical aspects of its characterization via subtomogram averaging).

- Line 297-301: Was there a particular location in the dolphin mouth where either RCU morphotype was found? Note that Simonsiella is typically on the palate (for example: <https://pubmed.ncbi.nlm.nih.gov/886011/>) and many oral microbes have a preferred habitat.

Samples collected in 2012 were obtained from the gingival sulcus.

Samples collected in 2018 were obtained from the palate, tongue, and gingival sulcus.

Samples collected in 2022 were obtained from the palate, buccal, and gingival sulcus.

We have added this information to the Methods section.

For samples collected in 2022, RBSs were observed in the palatal and gingival samples; no RBSs were observed in the buccal samples.

Beyond this, we did not pursue an investigation of the precise ecological habitat of RBS-As within the dolphin oral cavity.

- As mentioned above, FISH provides a straightforward route to identify the cells. Since FISH is working well for your cells, this is an obvious route. Haloquadratum provides a good example (e.g., <https://pubmed.ncbi.nlm.nih.gov/11207773/>)

We have performed additional FISH experiments. The results suggest that the RBSs are members of the Betaproteobacteria.

We did not pursue FISH at lower taxonomic levels as, to the best of our knowledge, there are no verified *Simonsiella*, Neisseriaceae, or Neisseriales probes available (as per ProbeBase). Designing and validating a *de novo* probe that is sufficiently specific and sensitive for a single taxonomic group seemed inefficient and unnecessary (see response to reviewer above).

- Figures: I would prefer the scale bars in figures to show the size of the bar so I could see cell size at a glance. Same comment for phase-contrast, FISH, and EMs.

In all microscopy figures, we have modified the scale bars to show the size of the bar in text.

We appreciate the comments and suggestions from this reviewer and believe that our analyses and manuscript have benefitted from them. We are happy to respond to any further questions and comments that you may have.

Reviewer #4 (Remarks to the Author):

The manuscript by Dudek and colleagues describes previously uncharacterized bacteria discovered in the mouth of bottlenose dolphins. The authors used phase contrast and fluorescence microscopy combined sequencing to suggest that the observed object is of bacterial origin and is not the part of the *Simonsiella* genus. Interestingly, the bacteria had rectangle-like shape which gave them the name RCU. Further investigations with cryo electron microscopy and tomography showed peculiar appendages which were commonly observed next to the other bacteria as well as periodic density covering the surface of the RCU.

While the features spark curiosity, the paper is surprisingly descriptive; the organism is unknown, the features of the lifestyle of RCUs are not described, the molecular identity of the appendages and of the regular pattern on the surface of the RCU is unknown. In the current form the paper is definitely more suitable for a more specialized journal.

We appreciate the reviewer's feedback and have carried out extensive new experiments to gain insight into the phylogenetic identity of the RCUs (now referred to as "RBSs" – see response to reviewer #2). FISH experiments suggest they are members of the Betaproteobacteria class. In addition, we have carried out new 16S rRNA gene amplicon sequencing experiments on 54 dolphin oral samples, 22 out of 48 (45.8%) of which were confirmed to contain RBSs via microscopy, to better characterize community composition. Spike-in experiments, where *S. muelleri* strain ATCC29453 cells were added directly to aliquots of dolphin oral samples that were previously sequenced and found to be *Simonsiella*-negative, served as valuable controls.

In addition to describing the RBSs, this work makes a valuable contribution in highlighting the morphological and ecological diversity hidden within so-called "microbial dark matter" – lineages of the tree of life for which no cultured representatives exist. As of 2016, 72% of the widely recognized bacterial phyla had no cultured representatives (Dudek et al., 2017, also see Hug et al., 2016). Furthermore, once their taxonomy is fully identified, the RBSs will provide an additional evolutionary case study for examining the genetic basis of bacterial multicellularity and longitudinal division (e.g., Nyongesa et al., 2022) and the evolutionary pressures selecting for such morphological and ecological features.

The study of uncultured bacteria has largely been restricted so far to metagenomic and single-cell genomics, in no small part due to the enormous challenges associated with characterizing these organisms when representatives

have not been cultured and/or attempts to do so have failed. Conversely, the study of cell biology has been largely limited to cells (including bacteria) that can be cultured, and thus there exists a severe bias in our understanding towards cells for whom laboratory conditions are conducive to growth.

While these challenges are substantial, we believe that in the absence of isolates, the complementation of genomics with other investigative modalities is imperative for characterizing the diversity of lifestyles across the tree of life. Our study provides a prime example of an unusual morphotype with novel appendages that would not be recognized through genomic methods alone, as the genomic underpinnings of these previously uncharacterized features are unknown.

We have modified the text to better emphasize the unique nature of our study in our introduction (paragraph #2).

Dudek, N.K., et al. "Novel microbial diversity and functional potential in the marine mammal oral microbiome." *Current biology* 27.24 (2017): 3752-3762.

Hug, L.A., et al. "A new view of the tree of life." *Nature microbiology* 1.5 (2016): 1-6.

Nyongesa, S., et al. "Evolution of longitudinal division in multicellular bacteria of the Neisseriaceae family." *Nature communications* 13.1 (2022): 1-18.

I was asked to comment on the cryo-EM part of the manuscript. Overall, it is well done with a few minor comments:

1. Scale bars in figure 5D,E don't seem to be 1 um long, please correct

Thank you for pointing this out. We have corrected the scale bar sizes in Figure 5D,E.

2. Tomograms in Figure 6 have sharp contrasty edges originating from tomographic reconstruction. They can be removed by normalizing the original tilt series to zero mean, softening their edges. Alternatively, as they don't contain useful information, the final images could be just cropped.

Thank you for drawing our attention to this matter. While edge artifacts from tomographic reconstruction such as these are to be expected, we agree that they

can be distracting and thus have cropped the images in Fig. 6 to improve their display as suggested.

3. Figure 7: panel D has inverted contrast while all the other images – not leading to representation inconsistencies.

Contrast shown for raw data is a matter of preference in cryoEM/ET and different features may be more clear with reversed contrast while for others the opposite may be true. In the case of this Figure, panel D shows line density plots; we think it is more intuitively logical for most people to see a “peak” where there is density as opposed to a “valley” where there is a gap. This required inverting the contrast of the micrographs since by default dense features in cryoEM/ET images have dark/negative contrast. The other images shown in prior Figures are either tomographic slices or 2D montages for which we performed no additional analyses that required inverting the contrast.

We appreciate the comments and suggestions from this reviewer and believe that our analyses and manuscript have benefitted from them. We are happy to respond to any further questions and comments that you may have.

Reviewer #1 (Remarks to the Author):

I am satisfied with the additional data presented and I feel the authors have addressed my initial comments. The paper is largely descriptive, which is ok considering the interesting phenotype of these microbes. Although a genome could not be definitively associated with the square-shaped bacteria, multiple lines of evidence point towards a putative identity. The authors are clear about this evidence and highlight important caveats in the interpretation.

Reviewer #3 (Remarks to the Author):

The authors did a lot of work to address comments from the initial reviews and I'm generally happy with everything. I do note a few minor problems that need attention before this work is published, as listed below. Overall, this is a fascinating study and I look forward to additional developments.

Minor points needing attention:

- I suggest writing the taxonomic rank first and the name second (e.g., genus *Stella*, family *Alcaligenaceae*, etc.). It's also recommended by the ICNP and other codes of nomenclature to italicize all formal taxonomic names, not just genus and species names (e.g., families, orders, etc.), although publishers often mess that up later in the process, which is ok.
- Please clean up the figure legend for Figure 4. In particular, the description of the section labeled "Criteria" needs work. I suggest describing the columns of this section in order from left to right and making sure the labels in the figure match the labels mentioned in the figure legend. I think (3) is missing in the current version.

Also, not related to the paper, but have you thought of grabbing RBS by micromanipulation or optical tweezers and trying to grow them in liquid or solid media? The growth conditions would still need to be right but this approach can work (e.g., <https://pubmed.ncbi.nlm.nih.gov/23950149/>, <https://pubmed.ncbi.nlm.nih.gov/7541115/>). I think Quake's lab still has an optical tweezer at Stanford.

- Brian Hedlund

Reviewer #4 (Remarks to the Author):

The updated manuscript by Dudek and colleagues is improved by a series of experiments suggesting that the bacteria imaged by the authors are distinct from the *Neisseriaceae* genera *Simonsiella* and *Conchiformibius* and are likely to be *Betaproteobacteria*. While it sheds more light on the potential organism that is being imaged, the origin and the functions of the observed features are still unknown.

Understanding that it is difficult to establish culturing and generic work for unknown species, I still believe that in the current form the manuscript is more suited to a more specialized journal.

Minor comments:

Lines 84 and 510: the resolution of tomograms is typically considered to be several nanometers; the current sentence suggests that it is one nanometer. Please rephrase.

Line 334: the reference to Figure 5C seems wrong, the image is not a tomogram.

Lines 349-352: the description of the thickness of the sample as a function of the tilting angle is trivial and in the opinion of the reviewer is not necessary here. Furthermore, the considerations below in this paragraph do not make sense – there may be some electron counts on the detector even for a very thick sample, but they will likely all correspond to multiple scattered electrons. I

don't see how any of these considerations lead to a conclusion that the sample is thinner than ~1.65 μm . In fact, if the authors want to make a statement about the thickness of the sample – they can measure it in tomograms or perform ice thickness measurements using the available methods, such as <https://pubmed.ncbi.nlm.nih.gov/29981485/>.

Figure 5d suggested that there is an OM layer between the presumably cytoplasmic volumes constricted by IM. The visualization in Figure 6c,f do not contain the OM layer, why is that?

Strangely, figure 7d,e is not mentioned in the text and is not discussed.

Finally, as a suggestion I would recommend the authors to apply a filtering technique such as non-linear anisotropic diffusion to reduce noise in their tomograms.

Reviewer #1 (Remarks to the Author):

I am satisfied with the additional data presented and I feel the authors have addressed my initial comments. The paper is largely descriptive, which is ok considering the interesting phenotype of these microbes. Although a genome could not be definitively associated with the square-shaped bacteria, multiple lines of evidence point towards a putative identity. The authors are clear about this evidence and highlight important caveats in the interpretation.

We thank the reviewer for their time and thoughtfulness in reviewing our manuscript. We believe the manuscript is substantially stronger as a result of their constructive feedback during the review process.

Reviewer #3 (Remarks to the Author):

The authors did a lot of work to address comments from the initial reviews and I'm generally happy with everything. I do note a few minor problems that need attention before this work is published, as listed below. Overall, this is a fascinating study and I look forward to additional developments.

We thank the reviewer for their time and thoughtfulness in reviewing our manuscript. We believe the manuscript is substantially stronger as a result of their constructive feedback during the review process. We too are excited for future studies that reveal additional insights into the biology of the RBSs!

Minor points needing attention:

- I suggest writing the taxonomic rank first and the name second (e.g., genus *Stella*, family Alcaligenaceae, etc.). It's also recommended by the ICNP and other codes of nomenclature to italicize all formal taxonomic names, not just genus and species names (e.g., families, orders, etc.), although publishers often mess that up later in the process, which is ok.

This is a helpful suggestion, thank you. We have revised the manuscript to ensure that taxonomic rank precedes name, where appropriate.

- Please clean up the figure legend for Figure 4. In particular, the description of the section labeled "Criteria" needs work. I suggest describing the columns of this section in order from left to right and making sure the labels in the figure match the labels mentioned in the figure legend. I think (3) is missing in the current version.

Thank you for highlighting this issue. We have updated the legend to reflect the revised Figure 4. Columns are described from left to right and matching column labels are mentioned in the legend. We have added a description for (3).

Also, not related to the paper, but have you thought of grabbing RBS by micromanipulation or optical tweezers and trying to grow them in liquid or solid media? The growth conditions would still need to be right but this approach can work (e.g., <https://pubmed.ncbi.nlm.nih.gov/23950149/>, <https://pubmed.ncbi.nlm.nih.gov/7541115/>). I think Quake's lab still has an optical tweezer at Stanford.

This is an interesting suggestion and something we will keep in mind for future efforts aimed at culturing the RBSs. Thanks!

- Brian Hedlund

Reviewer #4 (Remarks to the Author):

The updated manuscript by Dudek and colleagues is improved by a series of experiments suggesting that the bacteria imaged by the authors are distinct from the Neisseriaceae genera *Simonsiella* and *Conchiformibius* and are likely to be Betaproteobacteria. While it sheds more light on the potential organism that is being imaged, the origin and the functions of the observed features are still unknown.

Understanding that it is difficult to establish culturing and generic work for unknown species, I still believe that in the current form the manuscript is more suited to a more specialized journal.

We thank the reviewer for their time and thoughtfulness in reviewing our manuscript. We believe the manuscript is substantially stronger as a result of their constructive feedback.

While our manuscript is descriptive, we believe that its strength is in highlighting the fascinating and often under-recognized uncharacterized biological traits that have evolved within the uncultured majority of bacteria. These traits cannot be easily understood using commonly employed sequencing-based techniques (i.e., genome-resolved metagenomics or single-cell genomics) as their existence, let alone their genetic basis, has not been previously discovered. We hope our study encourages future researchers studying the uncultured majority of bacteria to pair their sequencing-based investigations with visual characterizations of cell morphology.

Minor comments:

Lines 84 and 510: the resolution of tomograms is typically considered to be several nanometers; the current sentence suggests that it is one nanometer. Please rephrase.

We have rephrased line 84 from:

"...allowed three dimensional (3D) imaging of intact bacterial cells at nanometer resolution20..."

to

"...allowed three dimensional (3D) imaging of intact bacterial cells at a resolution of a few nanometers20..."

And line 510 from:

"...enabled visualization of the structure of many RBS-A features at nanometer resolution."

to:

"...enabled visualization of the structure of many RBS-A features at close to nanometer resolution."

Line 334: the reference to Figure 5C seems wrong, the image is not a tomogram.

Thank you for catching this. We have moved the Fig. 5c reference to the immediately preceding sentence: "Dark, spherical structures were visible in the body of RBSs in cryoTEM images (Fig. 5c)."

Lines 349-352: the description of the thickness of the sample as a function of the tilting angle is trivial and in the opinion of the reviewer is not necessary here. Furthermore, the considerations below in this paragraph do not make sense – there may be some electron counts on the detector even for a very thick sample, but they will likely all correspond to multiple scattered electrons. I don't see how any of these considerations lead to a conclusion that the sample is thinner than ~1.65 um. In fact, if the authors want to make a statement about the thickness of the sample – they can measure it in tomograms or perform ice thickness measurements using the available methods, such as <https://pubmed.ncbi.nlm.nih.gov/29981485/>.

We indeed measured sample thickness at the RBS-A periphery directly from our tomograms, as reported in the sentence preceding lines 349-352.

The description of increasing thickness with increasing tilt angle may be useful for students and/or microbiologists with no experience in cryoET. However, we have eliminated it in favor of simplicity, at the Reviewer's suggestion.

We agree that increasing specimen thickness increases the probability of multiple scattering events. The limit for complete beam occlusion at 300 kV is ~1,650 nm, per the reference provided (45). Without tilting, our low-magnification 2D images and montages show some interpretable features through the entire RBS-As body such as putative lipid droplets and the membranes between segments. This indicates that the beam is not completely occluded by the RBS-As in the absence of tilt. Thus, taken together with Ref 45, we logically conclude that the RBS-As must be thinner than ~1,650 nm throughout. However, in favor of simplicity, we have deleted this conclusion.

We have changed the sentences in question from:

"Notably, tilting increases the mean free path (MFP) of electrons in the electron beam, that is, the apparent ice thickness through which they travel. This increase is in inverse proportion to the cosine of the tilt angle, resulting in up to 3-fold the thickness at 70° compared to that at 0°. However, the fact that the electron beam is not completely occluded in two-dimensional images without tilting, as seen in our cryoEM images, suggests that the thickness of the RBS-A body is under the ~1,650 nm thickness penetrance limit for the 300 kV acceleration voltage of the microscope we used to image the RBS-As, although useful information was deemed to be completely lost at thicknesses of 650 nm or greater in benchmarks using microcrystal electron diffraction."

to:

"Attempts to image the RBS-A body with cryoET failed because gold fiducials and cellular features quickly became indiscernible upon tilting, suggesting that the thickness of the specimen induced too many scattering events."

Figure 5d suggested that there is an OM layer between the presumably cytoplasmic volumes constricted by IM. The visualization in Figure 6c,f do not contain the OM layer, why is that?

As per the Fig. 6 caption, the OM layer is colored in green in the annotated examples in Fig. 6c,f: "...Blue, inner membranes; purple, periodic surface covering; green, outer membrane; red, matrix."

Strangely, figure 7d,e is not mentioned in the text and is not discussed.

Thank you for catching this. We now reference these subpanels of Fig. 7 explicitly in line 343:

"...manually measuring the distance between intensity peaks for 6-7 peaks (Fig. 7d,e)."

and in line 383:

"However, in our datasets, both the pilus-like appendages (Fig. 7a-c) and the S-layer-like surface feature (Fig. 7a,d,e) were structurally heterogenous..."

Finally, as a suggestion I would recommend the authors to apply a filtering technique such as non-linear anisotropic diffusion to reduce noise in their tomograms.

The tomograms are heavily filtered for annotation and display, being binned by 8x (this itself is stronger than NAD) and band-pass filtered (strong low-pass + mild high-pass filters) after SIRT-like filtration in IMOD during reconstruction. We tried NAD as well but there were no noticeable additional benefits.